# XMAP: Cross-population fine-mapping by leveraging genetic diversity and accounting for confounding bias

Mingxuan Cai [1] ✉, Zhiwei Wang [2,3], Jiashun Xiao[4], Xianghong Hu[2,3], Gang Chen[5,6,7] & Can Yang [2,3] ✉

Fine-mapping prioritizes risk variants identified by genome-wide association studies (GWASs), serving as a critical step to uncover biological mechanisms underlying complex traits. However, several major challenges still remain for existing fine-mapping methods. First, the strong linkage disequilibrium among variants can limit the statistical power and resolution of fine-mapping. Second, it is computationally expensive to simultaneously search for multiple causal variants. Third, the confounding bias hidden in GWAS summary statistics can produce spurious signals. To address these challenges, we develop a statistical method for cross-population fine-mapping (XMAP) by leveraging genetic diversity and accounting for confounding bias. By using cross-population GWAS summary statistics from global biobanks and genomic consortia, we show that XMAP can achieve greater statistical power, better control of false positive rate, and substantially higher computational efficiency for identifying multiple causal signals, compared to existing methods. Importantly, we show that the output of XMAP can be integrated with single-cell datasets, which greatly improves the interpretation of putative causal variants in their cellular context at single-cell resolution.

Genome-wide association studies (GWASs) have reported hundreds of thousands of associations between single-nucleotide polymorphisms (SNPs) and various phenotypes[1], but most reported SNPs reside in non-coding regions[2–4]. As the cell type and cellular process in which the identified SNPs are active remains largely unknown, the GWAS findings remain hard to interpret. Fine-mapping seeks to prioritize the causal SNPs underlying complex traits and diseases. Recent progress shows that, by integrating fine-mapping results and single-cell data, it becomes feasible to identify disease/trait-relevant cell types and cell states[5,6]. Therefore, fine-mapping is a critical step to interpret GWAS findings by elucidating their biological mechanisms of identified risk variants, and

fine-mapping results will offer an invaluable resource for precision medicine[7].

Despite the great promise of fine-mapping, efforts toward reliable prioritization of causal SNPs have been hampered by three key challenges. First, when GWAS samples come from a single population, SNPs in a local genomic region can be highly correlated due to the low recombination rates in that region. It is very difficult for statistical methods to distinguish the causal variants from a set of SNPs in strong linkage disequilibrium (LD). Second, genetic signals at trait-associated regions are commonly conferred by many variants acting together. A very recent study of 744 human expression quantitative trait loci (eQTLs) reported that 17.7% of the eQTLs harbour more than one

[1]Department of Biostatistics, City University of Hong Kong, Hong Kong SAR, China. [2]Guangzhou HKUST Fok Ying Tung Research Institute, Guangzhou 511458, China. [3]Department of Mathematics, The Hong Kong University of Science and Technology, Hong Kong SAR, China. [4]Shenzhen Research Institute of Big Data, Shenzhen 518172, China. [5]Hunan Provincial Key Lab on Bioinformatics, School of Computer Science and Engineering, Central South University, Changsha 410083, China. [6]WeGene, Shenzhen Zaozhidao Technology Co., Ltd, Shenzhen 518040, China. [7]Graduate Affairs, Faculty of Medicine, Chulalongkorn University, 10330 Bangkok, Thailand. ✉e-mail: mingxcai@cityu.edu.hk; macyang@ust.hk

variant with major effects on gene expression levels, emphasizing the importance of identifying multiple genetic variants within an associated locus[8,9]. For example, an eQTL associated with *ERPA2* and Crohn's disease was found to be driven by 13 separate variants[9]. However, it becomes computationally expensive to simultaneously search for multiple SNPs by enumerating causal combinations. Third, the unadjusted socioeconomic status[10] and geographic clustering[11,12] in GWAS samples can induce confounding bias in GWAS estimates[13]. These confounding factors cannot be fully corrected through linear mixed models (LMMs)[14,15] or principal component analysis (PCA)[16]. Fine-mapping without correcting the confounding bias in GWAS data can yield spurious results.

While many efforts have been devoted to the development of fine-mapping methods, existing methods only partially addressed the above major challenges. The classical fine-mapping methods[17,18] rely on an exhaustive search for all possible causal configurations of variants. They become computationally unaffordable when searching for more than three causal associations among thousands of variants. More efficient methods have been developed based on approximated inference, including CAVIARBF[19], FINEMAP[20], and DAP-G[21,22]. A very recent method, SuSiE[23,24], introduces a novel framework by assuming the overall genetic effects can be decomposed as a sum of single effects. The model structure of SuSiE enables an efficient algorithm to detect multiple causal SNPs with minor computational overhead. Its extension, SuSiE-inf[25], incorporates a polygenic component to improve reproducibility. Despite their improvement in computational efficiency, the statistical power of these methods is usually limited because it is difficult for them to distinguish the causal variants from the highly correlated variants in the single population setting. To boost the statistical power of fine-mapping, several methods were developed to leverage different LD patterns with cross-population GWASs, including trans-ethnic PAINTOR[26] and MsCAVIAR[27]. Although these methods allow a locus to harbour multiple causal variants in principle, they require enumerating all causal combinations of variants, hence become too time-consuming to search for more than three causal

variants. As a recent extension to SuSiE, SuSiEx[28] improves the computational efficiency of cross-population fine-mapping by inheriting the algorithm design from SuSiE. However, existing fine-mapping methods do not account for confounding bias in GWAS summary statistics, leading to spurious results.

In this paper, we develop a statistical method for cross-population fine-mapping (XMAP) by leveraging genetic diversity and accounting for confounding bias (Fig. 1). The success of XMAP relies on its three unique features. First, XMAP can leverage distinct LD structures from genetically diverged populations. It is known that individuals from different population backgrounds usually have different LD structures. For example, individuals from the African (AFR) population are known to have narrower LD compared to those from the European (EUR) population[29]. By jointly analyzing cross-population GWASs, XMAP can effectively improve the power and resolution of fine-mapping. Second, XMAP can identify multiple causal signals with a linear computational cost, while many existing fine-mapping methods are too time-consuming to identify multiple causal signals. Third, XMAP can correct the confounding bias in GWAS summary data to avoid false positive findings and improve reproducibility. Through comprehensive simulation studies, we show that XMAP not only improves the statistical accuracy of fine-mapping but also offers a substantial computational advantage over existing methods. The evidence from real data analysis indicates that XMAP achieves substantial power gain with high reproducibility. By combining the GWASs of low-density lipoprotein (LDL) from East Asian (EAS), African, and European, XMAP identifies three times more putative causal SNPs than SuSiE. These SNPs are strongly enriched in the eQTL of the liver, suggesting their important roles underlying the biological process of LDL. Furthermore, using the height GWAS as an example, we show that XMAP can effectively correct confounding bias and substantially improve reproducibility. Lastly but importantly, XMAP results can be integrated with single-cell data to identify trait-relevant cell populations at single-cell resolution, maximizing the utility of single-cell data for the inference of the pathological mechanisms. We apply XMAP to 12 blood

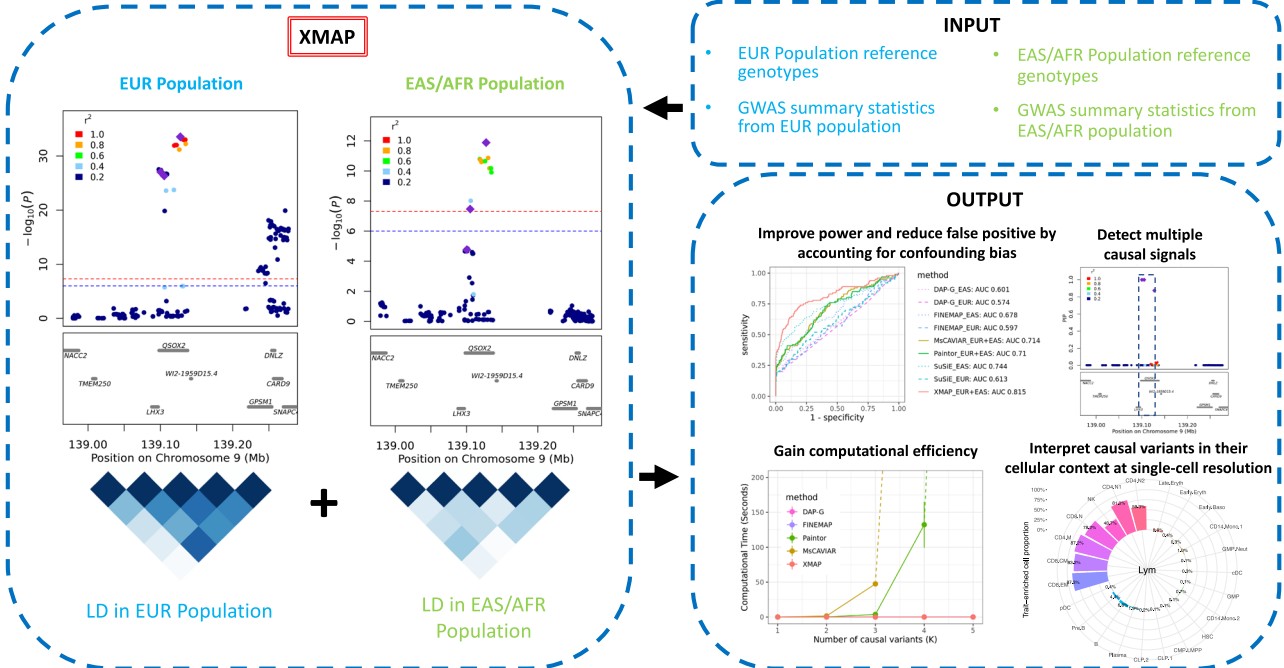

**Fig. 1 | XMAP overview.** XMAP takes the summary statistics and reference genotypes from multiple populations as inputs. XMAP can improve the statistical power of fine-mapping by leveraging the distinct LD pattern across populations while reducing false positives by accounting for confounding bias in GWAS summary statistics. Paired with a fast algorithm, XMAP is able to efficiently identify multiple causal signals. The fine-mapped SNPs can be integrated with single-cell datasets to identify trait-relevant cells.

traits and perform integrative analyses of the XMAP results and single-cell profiles of 23 hematopoietic cell populations. The analysis results suggest that XMAP enables the identification of the trait-relevant cell types in which putative causal SNPs are active. For example, SNPs identified by XMAP show a significant enrichment of the mean corpuscular volume in 99.3% of late-stage erythroid cells, which is very helpful to interpret GWAS results.

## Results

### Method overview

XMAP is a computationally efficient and statistically accurate method for fine-mapping causal variants using GWAS summary statistics. With innovations in its model and algorithm design, XMAP has three features: (i) It can better distinguish causal variants from a set of associated variants by leveraging different LD structures of genetically diverged populations. (ii) By jointly modeling SNPs with putative causal effects and polygenic effects, XMAP allows a linear-time computational cost to identify multiple causal variants, even in the presence of an over-specified number of causal variants. (iii) It further corrects confounding bias hidden in the GWAS summary statistics to reduce false positive findings and improve replication rates. The fine-mapping results given by XMAP can be further used for downstream analysis to illuminate the causal mechanisms at different cascades of biological processes, including tissues, cell populations, and individual cells. In particular, XMAP results can be effectively integrated with single-cell datasets to identify disease/trait-relevant cells. We provide the implementation of XMAP in an efficient and freely available R package at https://github.com/YangLabHKUST/XMAP. The technical details of XMAP are described in the Methods section.

### Simulation study

We conducted comprehensive simulation studies to compare the performance of XMAP with several related fine-mapping methods, including DAP-G, FINEMAP, SuSiE, SuSiE-inf, PAINTOR, MsCAVIAR, and SuSiEx. We also included an ad-hoc method that takes the union of the SNPs separately identified in different populations. This method is equivalent to the following procedure. First, we applied a single-population method to compute posterior inclusion probability (PIP) of SNP $j$ from different populations separately, denoted as $\text{PIP}_{j1},...,\text{PIP}_{jT}$, where $T$ is the number of populations. Then, for SNP $j$, its PIPs are compared across populations, and the largest one is selected to represent the 'merged PIP' of the SNP (i.e., $\text{PIP}_{j,merge} = \max\{\text{PIP}_{j1},...,\text{PIP}_{jT}\}$).

To mimic realistic LD patterns in different populations, we used genotypes of EUR samples from UKBB and genotypes of EAS samples from a Chinese cohort[30,31]. We considered a region between the base pair position 45,202,602 and 45,435,202 in chromosome 22 (GRCH37), which comprises $p = 500$ SNPs. To demonstrate the benefit of leveraging genetic diversity in different populations, we selected three candidate SNPs that satisfy the following properties: (i) In EUR population, they are in high LD (i.e., with absolute correlation > 0.9) with at least three non-causal SNPs. (ii) In EAS population, they are weakly correlated with non-causal SNPs (i.e., have an absolute correlation > 0.6 with less than two non-causal SNPs). The heat maps in Fig. 2b show the absolute correlation between the three candidate causal SNPs and their neighboring SNPs. We investigated $K_{true}$ causal SNPs, where $K_{true} \in \{1, 2, 3\}$, we randomly sampled $K_{true}$ from the three candidate SNPs as the causal ones. To mimic the unbalanced composition of GWAS samples in global populations, we considered $n_2 = 20,000$ samples from the EUR population and explored different sample sizes $n_1$ from the EAS population: 5000, 10,000, 15,000, and 20,000. For reference LD matrices, we used the EUR LD matrix estimated with 337,491 British UKBB samples provided in a recent study[32] and estimated the EAS LD matrix with 35,989 EAS samples from the Chinese cohort[30]. We designed our simulations in two scenarios. First, we

illustrated the benefit of cross-population fine-mapping by generating GWAS data without confounding bias. In the second scenario, we examined the effectiveness of XMAP in correcting confounding bias by simulating GWAS summary data with unadjusted sample structure.

We first consider the scenario in the absence of confounding bias. First, we generated the polygenic effects for all 44,728 SNPs in chromosome 22 which include the 500-SNPs target region. Specifically, we constructed the polygenic effects with $[\phi_{1j}, \phi_{2j}] \sim \mathcal{N}\left(\mathbf{0}, \begin{bmatrix} 0.005 & 0.004 \\ 0.004 & 0.005 \end{bmatrix}/500\right)$ for $j = 1,..., 44,728$, where 0.005 is the total heritability contributed by polygenic effects of the 500 SNPs in the target locus, with a per-SNP heritability $0.005/500 = 10^{-5}$ and a genetic correlation $\frac{0.004}{\sqrt{0.005 \times 0.005}} = 0.8$ between two populations. Then, we simulated the causal effects in the two populations with $\beta_{1k} \sim \mathcal{N}(0, \frac{0.25}{500})$ and $\beta_{2k} \sim \mathcal{N}(0, \frac{0.25}{500})$ for $k = 1,..., K_{true}$. This specification means that each causal SNP has a $0.25/0.005 = 50$ fold per-SNP heritability enrichment compared to non-causal SNPs, and the effect sizes of SNP $k$ are not necessarily the same across the two populations. The $K_{true}$ causal SNPs jointly contribute $0.25/500 \times K_{true} = 5 \times 10^{-4} \times K_{true}$ heritability. We obtained the standardized genotype matrices $\mathbf{X}_1 = [\mathbf{x}_{11},...,\mathbf{x}_{1p}] \in \mathbb{R}^{n_1 \times 44,728}$ and $\mathbf{X}_2 = [\mathbf{x}_{21},...,\mathbf{x}_{2p}] \in \mathbb{R}^{n_2 \times 44,728}$, whose columns have zero mean and unit variance. Given the genotypes and effect sizes, we generated quantitative phenotypes in the two populations with $\mathbf{y}_1 = \sum_{j=1}^{44728} \mathbf{x}_{1j}\phi_{1j} + \sum_{k=1}^{K_{true}} \mathbf{x}_{1[k]}\beta_{1k} + \mathbf{e}_2$ and $\mathbf{y}_2 = \sum_{j=1}^{44728} \mathbf{x}_{2j}\phi_{2j} + \sum_{k=1}^{K_{true}} \mathbf{x}_{2[k]}\beta_{2k} + \mathbf{e}_2$, where $\mathbf{x}_{1[k]}$ and $\mathbf{x}_{2[k]}$ are the columns of $\mathbf{X}_1$ and $\mathbf{X}_2$ corresponding to the $k$-th causal SNP, and $\mathbf{e}_1 \sim \mathcal{N}(\mathbf{0}, (1 - 0.005 - 5 \times 10^{-4} \times K_{true})\mathbf{I}_{n_1})$ and $\mathbf{e}_2 \sim \mathcal{N}(\mathbf{0}, (1 - 0.005 - 5 \times 10^{-4} \times K_{true})\mathbf{I}_{n_2})$ are independent noise in the two populations, respectively. Finally, we computed the GWAS summary statistics by marginally regressing the simulated phenotypes on each SNP for each population (Fig. 2a). For XMAP, we first used the summary statistics of all 44,728 SNPs to estimate the polygenic parameters and inflation constants. Then, we ran variational EM algorithm with the 500 SNPs in the target region. For other methods, we directly applied them to the 500 SNPs in the target region. The details of data pre-processing and parameter settings of XMAP and compared methods are given in the Supplementary Note. For each setting, we generated 50 replicates and identified causal variants by controlling the global false discovery rate (FDR). To control the global FDR, we first compute the local FDR of each SNP as $fdr_j = 1 - \text{PIP}_j$. Then we sort SNPs by local FDR in ascending order and regard the $j$-th re-ordered SNP as a risk SNP if $FDR_{(j)} = \frac{\sum_{i=1}^{j} fdr_{(i)}}{j} < \xi$, where $fdr_{(i)}$ is the $i$-th ordered local FDR, $FDR_{(j)}$ is the corresponding global FDR, and $\xi$ is the selected threshold to control the global FDR (e.g., $\xi = 0.1$). We identified putative causal SNPs with a given FDR level and computed the empirical FDR as $1 - \frac{\text{NO. of true SNPs among putative causal SNPs}}{\text{NO. of putative causal SNPs}}$. Because a locus includes at most 3 causal variants in our simulation, we aggregated 50 simulation replicates to improve the precision of empirical FDR.

We first evaluated the FDR calibration of compared methods. Figure 2c shows the expected FDR and empirical FDR when $K_{true} = 3$ and $n_1 = n_2 = 20,000$. As expected, XMAP had well-controlled FDR across all thresholds. By contrast, without accounting for the polygenic effects, SuSiEx and XMAP with $\mathbf{\Omega} = \mathbf{0}$ are inflated with a very similar pattern, suggesting they produced many false positives. To better understand these results, we compared the variance of causal effects ($\sigma_{k1}^2$ and $\sigma_{k2}^2$ for $k = 1,...,5$) estimated by XMAP and XMAP ($\mathbf{\Omega} = \mathbf{0}$), which is equivalent to SuSiE when there is no confounding bias. We did not include SuSiEx in this comparison because it does not output $\hat{\sigma}_{k1}^2$ and $\hat{\sigma}_{k2}^2$. However, we expect XMAP ($\mathbf{\Omega} = \mathbf{0}$) to have similar parameter estimates with SuSiE because they have the same statistical model and similar calibration performance. For ease of

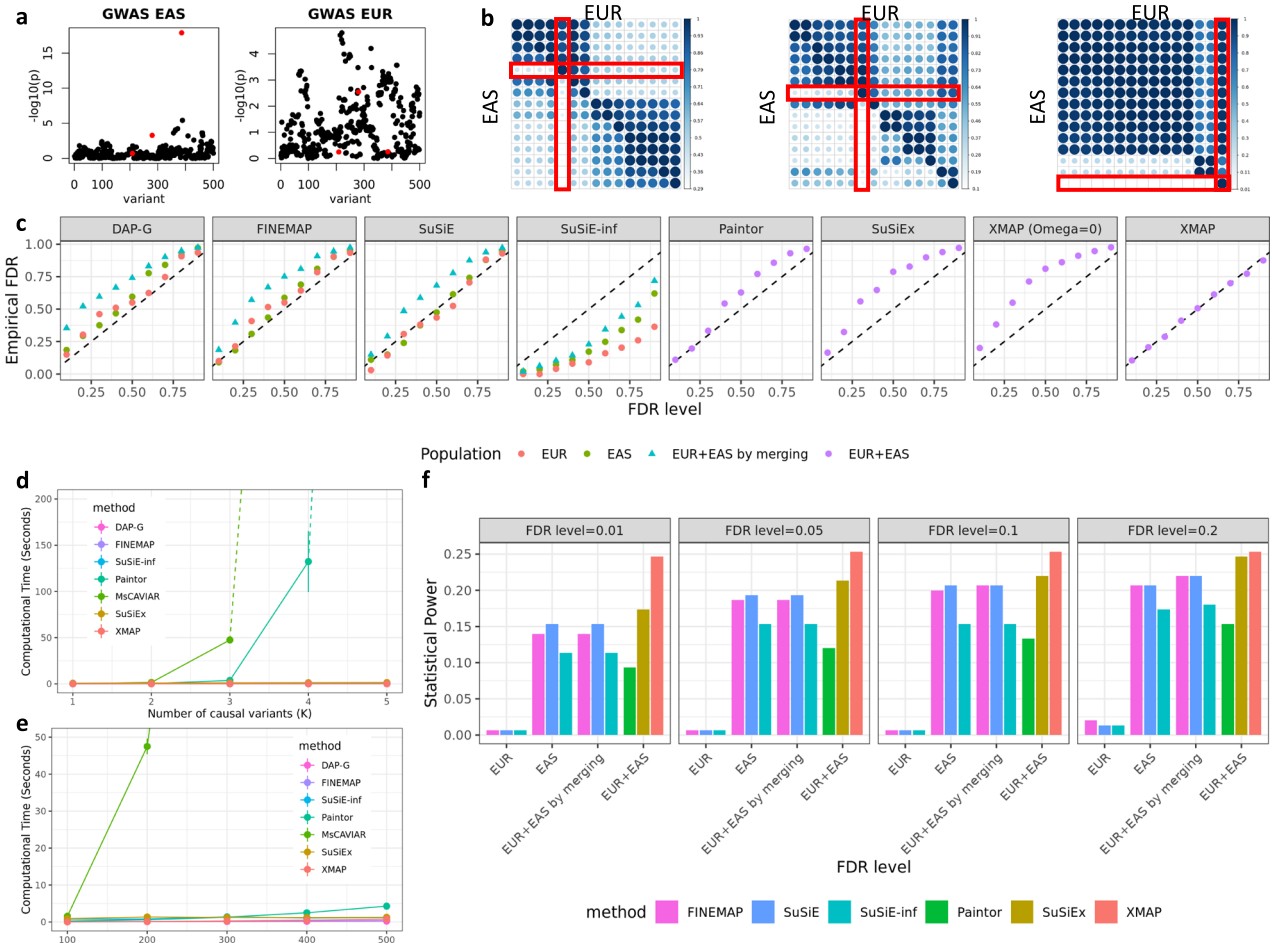

**Fig. 2 | Comparisons of fine-mapping approaches in the absence of confounding bias. a** Manhattan plots of a simulated GWAS data in EAS (left) and EUR (right). All *p*-values are obtained from marginal regression. **b** Heat maps showing the absolute correlations between the three causal SNPs (highlighted with rectangles) and their nearby SNPs in EAS and EUR populations. **c** Comparisons of FDR control with $n_1 = n_2 = 20,000$ and $K_{true} = 3$. We include an ad-hoc method by merging the discoveries from EAS and EUR (cyan triangles), which is equivalent to controlling the FDR by using the largest PIP across populations. This post-selection procedure introduces false positives. By contrast, XMAP is well-calibrated by

modeling cross-population GWASs through an integrated statistical framework. **d** CPU timings of XMAP, MsCAVIAR, PAINTOR, FINEMAP, and DAP-G are shown for increasing $K$ with $p = 100$. Solid lines are CPU time recorded in our experiments and dashed lines represent predicted CPU time based on the time complexity of corresponding approaches. **e** CPU timings are shown for increasing $p$ with $K = 2$. **f** Comparisons of statistical power with $n_1 = n_2 = 20,000$ and $K_{true} = 3$. Computational time is presented as the mean value +/- standard deviation. Results and error bars are summarized from 50 replications.

demonstration, we considered the case of only one causal effect, i.e., $K_{true} = 1$, and sorted $\hat{\sigma}_{k1}^2$ and $\hat{\sigma}_{k2}^2$ in decreasing order, respectively. As such, the true causal effect with variance 0.25/500 should be captured by the first causal component in the model. The remaining four causal components in the model are redundant with true variance of zero. As observed in Supplementary Figure 10, although here we set $K = 5 > K_{true}$, XMAP produced unbiased estimate of $\sigma_{k1}^2$ and $\sigma_{k2}^2$ by taking the polygenic effects into account. In contrast, without modeling the polygenic effects, XMAP ($\Omega = 0$) overestimated $\sigma_{k1}$ and $\sigma_{k2}$, making its PIP inflated. PAINTOR had a satisfactory performance with stringent FDR thresholds (FDR ≤ 0.3) but became inflated with larger FDR thresholds. The ad-hoc method by merging discoveries from a single-population method produced inflated PIP because this procedure involves a step of taking the maximum of PIP obtained from multiple populations. Indeed, this post-selection step introduces selection bias and inflates false positive findings. Among the single-population methods, FINEMAP and SuSiE had satisfactory FDR with stringent thresholds. Although SuSiE-inf can account for the polygenic effects, it was conservative because it only used GWAS from a single population. Furthermore, compared to SuSiE-inf, XMAP produced more stable

estimates of polygenic parameters (Supplementary Fig. 10) by taking advantage of its algorithm design and leveraging the information of the entire chromosome.

Next, we evaluated the statistical power of compared methods. We excluded DAP-G in this comparison because it had inflated FDR. As shown in Fig. 2f, XMAP was clearly the overall winner with the highest statistical power across different FDR thresholds. It had the largest gain when the FDR was controlled at a stringent level (e.g. FDR=0.01). Because MsCAVIAR was too time-consuming to handle more than two causal variants, we only applied MsCAVIAR to the setting with $K_{true} \in \{1, 2\}$. We provide the full results in Supplementary Figs. 1–3. These results indicate that XMAP is not only well-calibrated in the existence of polygenic effects but also achieve higher power than existing methods. To further investigate the difference in fine-mapping performance, we contrasted the PIP obtained by XMAP with those obtained by other methods (Supplementary Figs. 7–9). Clearly, XMAP produced substantially higher PIP for causal variants, as compared to other methods, suggesting that XMAP could better distinguish causal SNPs from non-causal SNPs. This explains our observation that XMAP often yields higher statistical power. We also

assessed the resolution of fine-mapping by evaluating the size of credible sets. The smaller credible sets, the higher resolution of fine-mapping. Here we consider XMAP, FINEMAP, SuSiEx, SuSiE-inf, and SuSiE because they are the only methods that can provide credible sets for individual causal signals. As summarized in Supplementary Fig. 12, XMAP was the only method that could produce level-95% credible sets with a median size of one in all settings. We also assessed the sensitivity and specificity under various PIP thresholds and summarized the partial area under the receiver operating characteristic curve (pAUC) with false positive rate (FPR) thresholds 0.1, 0.2, and 0.3 (Supplementary Figs. 7–9). With FPR thresholds of 0.2, and 0.3, XMAP had the best performance among cross-population methods when the EAS sample size was large ($n_2 = 20{,}000$). When FPR = 0.1, all methods had similar pAUC. It is important to note that pAUC only represents the relative rankings of true causal SNPs under an empirical FPR threshold. A method can have a high pAUC but low power if the PIP of all SNPs are small. For example, although SuSiE-inf had the highest pAUC, its statistical power was much lower than XMAP. We used $K = 5$ for XMAP in the main results and investigated $K = 10$ in the Supplementary Figs. 1–3. Under both settings, XMAP had consistent performance and steadily outperformed compared methods, suggesting its robustness to the specification of $K$. More comparisons under different settings of $n_1$, $n_2$ and $K_{true}$ are provided in the Supplementary Figs. 1–9.

To investigate the computational efficiency, we evaluated the CPU time of compared methods under different settings of $K$ and $p$. As shown in Fig. 2d, the computational cost of MsCAVIAR and PAINTOR increases exponentially with both $K$ and $p$. When analyzing a locus with $p = 100$ SNPs, MsCAVIAR could only include $K \leq 4$ causal signals and PAINTOR could only include $K \leq 5$ causal signals. It took more than one week for them to finish the analysis when more signals were included. By contrast, the computational cost of XMAP is linear to $K$, which makes it highly efficient when applied to a locus with multiple causal SNPs. To identify multiple causal signals, the computational efficiency of XMAP allows us to set $K$ to a large value (e.g., $K = 10$) when $K_{true}$ is unknown. While SuSiE-inf, SuSiEx, DAP-G, and FINEMAP had CPU times comparable to XMAP, they could not simultaneously leverage cross-population GWASs and account for polygenic effects to improve fine-mapping. This benchmark was evaluated using a Linux computing platform with 20 CPU cores of Intel (R) Xeon (R) Gold 6152 CPU at 2.10 GHz processor.

To assess the robustness of XMAP when the genetic effects are different from the model assumption, we include two sets of additional simulation that consider the absence of polygenic effects and misspecified genetic effects. In the setting without polygenic effects (Supplementary Figs. 23–30), XMAP still had comparable performance with existing cross-population methods because we allow the polygenic effects to be adaptively estimated from the data. To mimic the scenario when the XMAP assumptions on effect sizes is violated, we conducted simulation by generating effect sizes from scaled $t$-distributions with degrees of freedom 16 and 4 (see Supplementary Note), representing different levels of discrepancies from the normal distribution. Overall, XMAP performed reasonably well in controlling false positives while achieving high statistical power when the effect size distribution is misspecified (Supplementary Figs. 13 and 14).

In the second set of simulations, we focus on fine-mapping of GWAS data in the presence of uncorrected confounding bias. We introduced sample structures to GWAS data by using the genotype principal components following a previous work[33]. Specifically, we first performed PCA on the genotypes of EAS and EUR samples separately and extracted the first principal components (PCs) from the two populations as representations of sample structures, denoted as $PC_1 \in \mathbb{R}^{n_1}$ and $PC_2 \in \mathbb{R}^{n_2}$, respectively. We re-scaled $PC_1$ to have mean zero and variance 0.05 and re-scaled $PC_2$ to have mean zero and variance 0.2. These variance values were selected to introduce the proper level of inflation in the summary statistics. Next, we generated

quantitative phenotypes with $\mathbf{y}_1 = PC_1 + \sum_{k=1}^{K_{true}} \mathbf{x}_{1[k]}\beta_{1k} + \mathbf{e}_1$ and $\mathbf{y}_2 = PC_2 + \sum_{k=1}^{K_{true}} \mathbf{x}_{2[k]}\beta_{2k} + \mathbf{e}_2$, where the generating distributions of $\beta_{1k}$ and $\beta_{2k}$ are the same as those in the first scenario and the independent errors were generated with $\mathbf{e}_1 \sim \mathcal{N}(\mathbf{0}, (1 - 5 \times 10^{-4} \times K_{true} - 0.05)\mathbf{I}_{n_1})$ and $\mathbf{e}_2 \sim \mathcal{N}(\mathbf{0}, (1 - 5 \times 10^{-4} \times K_{true} - 0.2)\mathbf{I}_{n_2})$. Finally, we simulated two sets of GWAS summary data. The first set of GWAS data was obtained by regressing phenotype vectors on each SNP without including the PCs as covariates, representing the scenario with unadjusted confounding bias. We constructed the second set of GWAS data with confounding bias adjusted by performing marginal regression while including the PCs as covariates. Figure 3b shows the inflation constants in the simulated GWASs with confounding bias, evaluated by estimated LDSC intercepts $\hat{c}_1$ and $\hat{c}_2$. The inflation constants were substantially larger than one, indicating a strong confounding bias. The confounding bias became stronger when the sample size increased, suggesting an exacerbated inflation in GWAS summary statistics. To evaluate the performance of XMAP's adjustment of confounding bias, we applied XMAP and compared methods to both sets of GWAS summary statistics. As we can observe in Fig. 3a and Supplementary Fig. 16, all existing methods except PAINTOR produced well-calibrated PIP when the PCs were corrected in the GWAS summary data, suggesting the importance of including PC in the association mapping stage before fine-mapping. When the PCs were not corrected, all existing methods had different levels of FDR inflation, suggesting that they requires correcting confounding factors in the association mapping stage before conducting fine-mapping. However, in real GWAS data, many confounding effects, such as socioeconomic status[10] and geographic clustering[11,12] in GWAS samples, cannot be fully corrected by including PCs as covariates. By contrast, XMAP was the only method that could still achieve satisfactory FDR control performance by effectively correcting the confounding bias in GWAS data when the confounding effects are not adjusted in the summary data. We summarized the receiver operating characteristic (ROC) curve of compared methods in Fig. 3c. When the PCs were corrected in GWAS summary data, XMAP had an area under the curve (AUC) of 0.863 when the EAS cohort had a sample size of 5,000, which we considered as the highest achievable AUCs. When the PCs were not corrected in GWAS summary data, XMAP could still achieve 90.8% (0.784/0.863) of the optimal AUC. As a comparison, MsCAVIAR only achieved 88.6% (0.765/0.863) of the optimal AUC. In other settings (Supplementary Figs. 20–22), the AUC of XMAP was also comparable with other cross-population methods. This evidence suggests that XMAP can effectively reduces the false positive findings under the LDSC assumptions, making it more reliable than existing methods in the presence of confounding bias. Here, we showed a concrete example with a single causal signal in Fig. 3d as an illustration. With uncorrected confounding bias, the GWAS $p$-values were inflated in the left regions of the locus (first column of Fig. 3d). Without accounting for the confounding bias, SuSiE, SuSiE-inf, and SuSiEx produced a false positive signal (SNPs in green circles in the corresponding panels of Fig. 3d) and assigned a high PIP $\approx 0.6$ for a null SNP. By adjusting the estimation error of GWAS effects based on inflation constants $\hat{c}_1$ and $\hat{c}_2$, XMAP effectively reduced the PIP of SNPs related to the false positive signal and correctly excluded the false positive signal from level-95% credible sets (panel with the label 'XMAP' in Fig. 3d). When we forced XMAP to ignore the inflation by setting $\hat{c}_1 = \hat{c}_2 = 1$, the false positive signal appeared in the output (panel with the label 'XMAP (C=I)' in Fig. 3 d), indicating the confounding bias was not properly adjusted. This observation implies the effectiveness of using the inflation constants to correct confounding bias in GWAS.

## Real data analysis
We performed fine-mapping to identify putative causal SNPs of complex traits with cross-population GWASs. First, by applying XMAP to LDL GWASs, where the magnitude of confounding bias was ignorable, we illustrated XMAP's superior performance in improving fine-

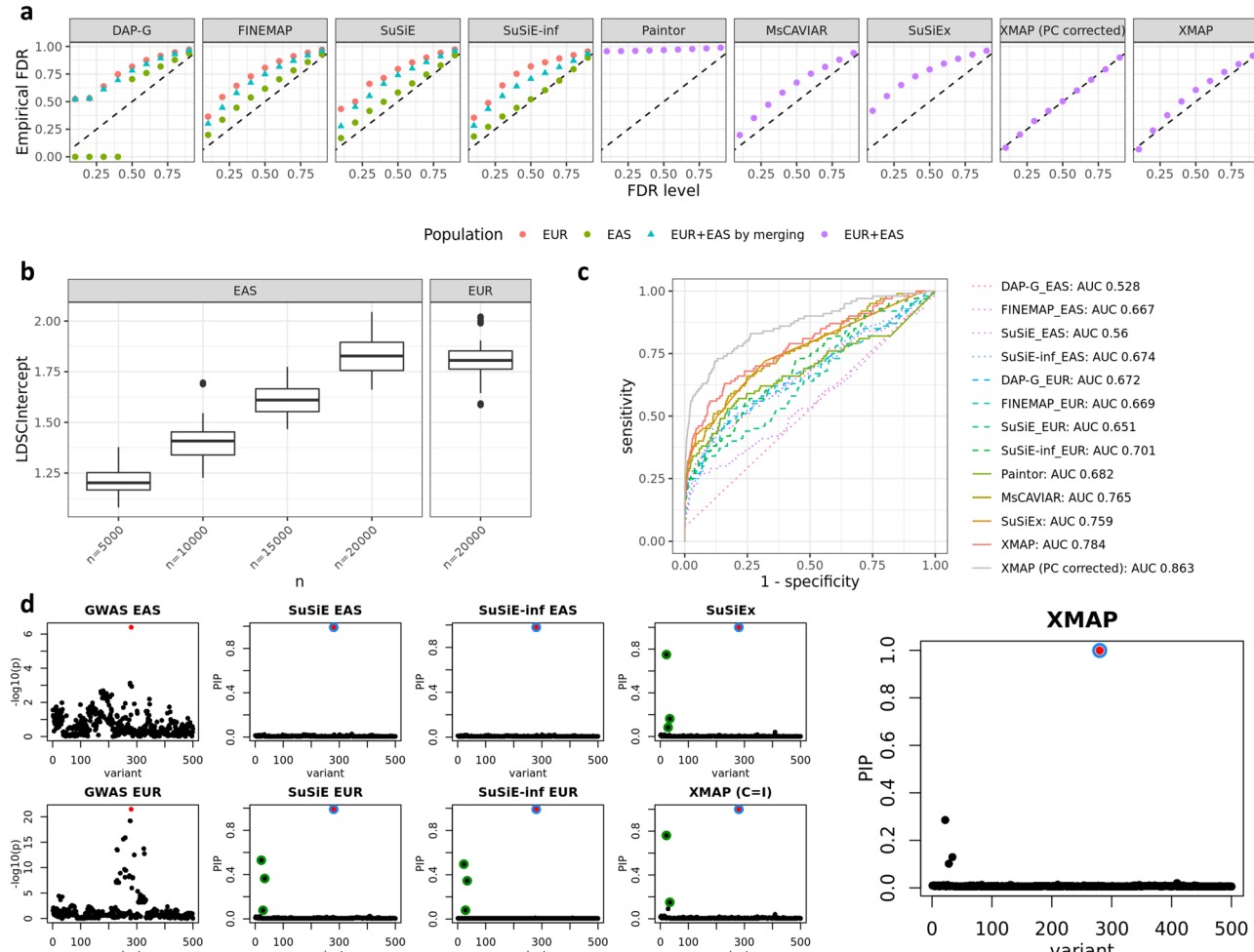

**Fig. 3 | Comparisons of fine-mapping approaches in the presence of confounding bias. a** Comparison of FDR control. **b** Estimated LDSC intercepts $\hat{c}_1$ (EAS) and $\hat{c}_2$ (EUR) with sample size $n_2 = 20,000$ in EUR and $n_1 \in \{5,000, 10,000, 15,000, 20,000\}$ in EAS. Box plot bounds show the lower, median, and upper quartiles; whisker lengths are 1.5 times the interquartile range; and points beyond the whiskers are outliers. **c** ROC curves of XMAP, PAINTOR, MsCAVIAR, SuSiE, FINEMAP, and DAP-G with $K_{true} = 1$, $n_1 = 5000$, $n_2 = 20,000$. **d** An illustrative example generated by simulation. The first column shows the $-\log_{10}(p)$-value in the GWAS of EAS(top) and EUR (bottom) obtained from marginal

regression without correcting the PC. The second column shows the PIP obtained by applying SuSiE to the training data of EAS (top) and EUR (bottom). The third column shows the PIP obtained by applying SuSiE-inf to the training data of EAS (top) and EUR (bottom). The fourth column shows the PIP obtained from SuSiEx (top) and XMAP by setting $\hat{c}_1 = \hat{c}_2 = 1$ (bottom). The last panel shows the PIP obtained from XMAP with $c_1$ and $c_2$ estimated from the data. Red dots represent causal SNPs. Circles in the same color represent SNPs in the level-95% credible sets of a causal signal. Results are summarized from 50 replications.

mapping power and resolution. Second, to investigate the ability of XMAP in correcting confounding bias, we applied XMAP to combine height GWASs from an EAS cohort[30] and the British cohort in UKBB, which was known to be affected by population structure[11,12]. Through replication analysis, we compared the credibility of XMAP fine-mapped SNPs with related methods. Third, with the confounding bias properly corrected, we showed that XMAP enables the identification of multiple causal signals within a locus. Last but importantly, we integrated the fine-mapping output of XMAP in blood traits with single-cell data. With the improved fine-mapping results, we can have a better interpretation of risk variants in their relevant cellular context, gaining biological insights into causal mechanisms at single-cell resolution.

## XMAP improves fine-mapping by leveraging genetic diversity

We first applied XMAP to analyze LDL by combining GWASs from EUR, EAS, and AFR. As discovery cohorts, we used the GWASs of AFR and EAS released by the Global Lipids Genetics Consortium (GLGC)[34], which were obtained based on 92,934 AFR samples and 71,150 EAS samples, respectively. For EUR, we considered two GWAS datasets: the UKBB GWAS summary data released by the Neale Lab with a sample

size of 343,621 (http://www.nealelab.is/uk-biobank/), and the EUR GWAS data from GLGC with a sample size of 664,450[34]. These GWAS summary statistics included 11,569,928-35,328,891 genotyped and imputed autosomal SNPs, minimizing the risk of omitting causal variants. Details of GWAS summary statistics are summarized in Supplementary Table 1. For EAS and EUR, we used the same reference LD matrices as in our simulation studies. For AFR, we estimated the LD matrices by using 3,072 African individuals from UKBB as reference samples. We followed a previous work[32] to partition all autosomal chromosomes into 2763 consecutive loci, each with a width of 1 million base pairs (Mbp). To fully account for LD when analyzing each 1 Mbp locus, we included all SNPs in an extended region that also covers 1 Mbp before the starting position and 1 Mbp beyond the ending position of the locus, leading to a 3 Mbp extended region. We excluded the MHC region (25.5 Mbp–33.5 Mbp in chromosome 6) and two other long-range LD regions (8 Mbp–12 Mbp in chromosome 8 and 46 Mbp–57 Mbp in chromosome 11) because many spurious results were reported in these regions[32]. We applied XMAP to all regions that have more than 100 SNPs after overlapping the reference LD matrices with GWAS data. Because SuSiE, SuSiE-inf, and SuSiEx achieved the

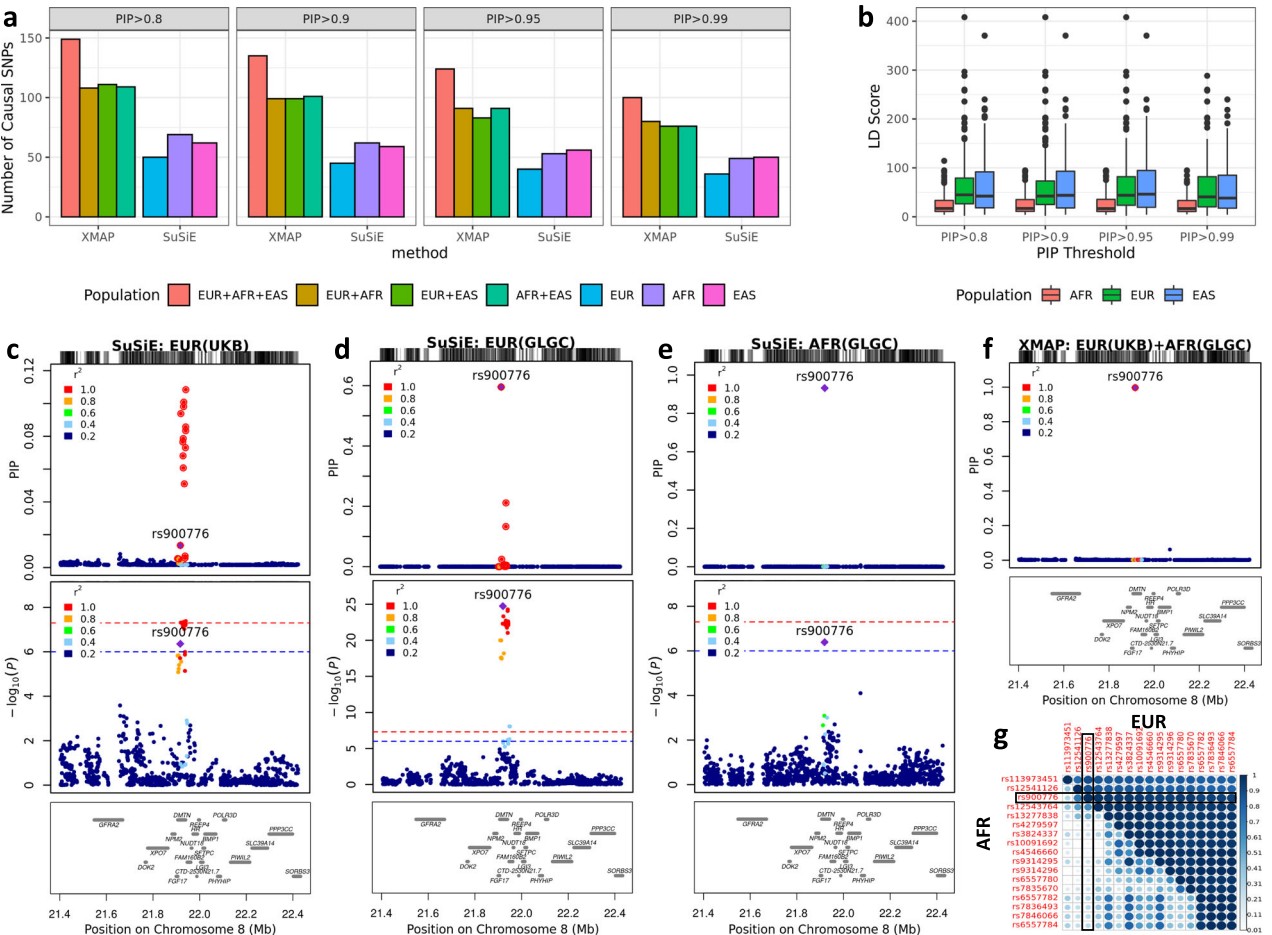

**Fig. 4 | Application of XMAP and SuSiE to LDL GWAS from EUR ($n$ = 343,621), AFR ($n$ = 92,934), and EAS ($n$ = 71,150). a** The number of causal signals identified by XMAP and SuSiE with PIP thresholds 0.8, 0.9, 0.95, and 0.99. Colors represent different combinations of GWAS training data. **b** Box showing the LD score distribution of putative causal SNPs identified by XMAP. Box plot bounds show the lower, median, and upper quartiles; whisker lengths are 1.5 times the interquartile range; and points beyond the whiskers are outliers. **c–f** Fine-mapping of locus 21.4 Mbp–22.4 Mbp in chromosome 8. The fine-mapping methods and training data

are labeled on top of each panel. The top panels show the PIP. SNPs within the 99% credible set are highlighted with red circles. Middle panels show the $-\log_{10}(p-\text{value})$ in GWAS. Red dashed lines represent $5 \times 10^{-8}$. Blue dashed lines represent $1 \times 10^{-6}$. The bottom panels annotate the position of genes in the locus. **g** Absolute correlation in EUR and AFR among the SNPs within the level-99% credible set as shown in the red circles of (**c**). The SNP rs900776 is highlighted in the heat map.

best performance among existing methods in our simulation studies, we applied them to the GWAS of LDL, serving as a baselines for comparison. We set $K = 10$ in all compared methods.

We first quantified the confounding bias in these GWAS data using the estimates of LDSC intercepts. As shown in Supplementary Table 1, the LDSC intercepts estimated from all LDL GWASs were not substantially different from one, suggesting an ignorable confounding bias here. We then summarized the fine-mapped SNPs in Fig. 4a. By combining GWAS data from different populations, XMAP consistently identified more causal signals than SuSiE with different PIP thresholds. Specifically, XMAP identified 149 SNPs with PIP > 0.8 and 145 SNPs with PIP > 0.9 when the GWASs from all three populations were jointly analyzed, which was three times more than the number of SNPs identified by SuSiE in EUR (50 SNPs with PIP > 0.8 and 45 SNPs with PIP > 0.9). To assess the credibility of the putative causal SNPs, we evaluated the replication rate using an independent LDL GWAS[34] from the EUR population with a median sample size of 85785 (see Supplementary Table 1). We computed PIP in this replication cohort using SuSiE and summarized the replication performance in Supplementary Fig. 31. Clearly, XMAP was the overall winner with the highest replication rate under various PIP thresholds. As expected, directly merging SuSiE discoveries across single population analyses (denoted as SuSiE-

merge) had a much lower replication rate than XMAP. For example, when GWASs from all three populations were combined, with a PIP threshold of 0.8, 25.5% (38/149) SNPs identified by XMAP had PIP > 0.1 in the replication cohort. With the same PIP threshold, only 18.9% (32/109) SNPs identified by the merging procedure had PIP > 0.1 in the replication cohort. With a more stringent PIP of 0.99, 31% (31/100) SNPs identified by XMAP had replication PIP > 0.1. As a comparison, there were only 18.9% (24/127) SNPs identified by the merging procedure that had PIP > 0.1. Therefore, directly merging the findings of single-population methods is not an optimal way to combine cross-population GWASs because it can introduce post-selection bias. The complete fine-mapping results are available at https://github.com/YangLabHKUST/XMAP/tree/main/results.

The improved statistical power of XMAP could be attributed to its capacity of leveraging genetic diversity. To see this, we checked the LD score which is a summation of the squared correlation between a SNP and other SNPs in a population. A large LD score of a SNP means that this SNP has strong LD with many other SNPs. We observed that the XMAP fine-mapped SNPs have the smallest LD scores in AFR (Fig. 4c), suggesting the power gain of XMAP could be attributed to the weak LD between causal SNPs and non-causal SNPs in AFR. As an example, rs900776 is an intronic variant in the *DMTN* region, which is highly

correlated with surrounding SNPs in EUR. Because of this, SuSiE estimated the PIP of rs900776 as small as 0.002 using UKBB GWAS and produced a very large 99% credible set that included 16 other SNPs for this signal. When applying SuSiE to the larger EUR GWAS data from GLGC, the PIP of SNP rs900776 increased from 0.002 to 0.6 (Fig. 4d). Different from the LD pattern in European population, rs900776 is less correlated with nearby SNPs in African population (Fig. 4g). Therefore, when SuSiE was applied to AFR GWAS, the estimated PIP of rs900776 increased to 0.9 (Fig. 4e). Unlike SuSiE which analyzes a single population at a time, XMAP enables joint analysis of EUR and AFR GWASs. XMAP successfully identified SNP rs900776 with a PIP as high as 0.99, yielding a high-resolution credible set which contains rs900776 only (Fig. 4f). This indicates the improved power and resolution of XMAP by leveraging genetic diversity.

## XMAP enables the correction of confounding bias in fine-mapping

To demonstrate the effectiveness of XMAP in the correction of confounding bias, we applied XMAP to the height GWASs which were well known to be affected by population structure[11,12]. Following the previous cross-population fine-mapping pipeline[35], we first applied fine-mapping methods to discovery GWAS datasets, and then evaluated the credibility of fine-mapped SNPs in replication datasets from different population backgrounds. Here, we used the EUR GWAS from UKBB and a Chinese GWAS in our previous study[30] as discovery cohorts. For replication, we considered a recently released within-sibship GWAS from European population, which was known to be less confounded by population structure. We also included the GWAS from BBJ cohort as replication data from EAS background. To ensure the SNP density, these GWASs were imputed to cover 3,776,576–12,515,778 variants (see Supplementary Table 1). The LDSC intercepts of UKBB GWAS and BBJ GWAS were estimated as 1.66 (s.e. = 0.042) and 1.39 (s.e. = 0.024), respectively, indicating the presence of strong confounding bias. The LDSC intercepts of EUR Sibship GWAS and Chinese GWAS were estimated as 1.07 (s.e. = 0.0089) and 1.12 (s.e. = 0.012), suggesting that the confounding bias is nearly ignorable.

We summarized the replication rates of fine-mapped SNPs in Fig. 5. Overall, XMAP had the best replication performance. Among the overlapped SNPs between the EUR Sibship GWAS and discovery cohorts, SuSiE detected 306 SNPs with PIP > 0.8 from UKBB GWAS. However, only 14.1% (43/306, Fig. 5a) were found to be genome-wide significant and only 13.4% of them (41/306, Fig. 5b) had PIP > 0.1 in the EUR Sibship replication cohort. The low replication rate suggests that these SNPs could be false positive signals due to unadjusted confounding bias. Consistent with our observation in the analysis of LDL GWAS, the replication rate of SuSiE-merge was even lower because it introduced post-selection bias. Although SuSiEx identified more SNPs by incorporating cross-population GWASs, a large proportion of them could not be replicated. By adjusting the confounding bias, XMAP successfully reduced the number of false positive signals. For example, using PIP > 0.8 as a threshold, 21.4% (44/206) SNPs detected by XMAP were genome-wide significant and 21.4% (44/206) had PIP > 0.1 in EUR Sibship replication cohort. A similar pattern can be observed in the BBJ replication cohort. With a PIP threshold of 0.8, only 23.9% (54/226, Fig. 5c) SNPs detected from UKBB GWAS by SuSiE were genome-wide significant and 8.8% (19/226, Fig. 5d) had PIP > 0.1 in BBJ GWAS. As a comparison, 42.3% (71/168) SNPs detected by XMAP were genome-wide significant and 14.9% (25/168) had PIP > 0.1 in BBJ replication cohort. The higher replication rate of XMAP implies its effectiveness of fine-mapping by accounting for confounding bias.

Although PAINTOR and MsCAVIAR can also integrate cross-population GWASs, they are too time-consuming to analyze all loci on the genome. Here, we consider a concrete example to compare the performance of cross-population methods in the presence of

confounding bias (Fig. 5). For XMAP, we considered two settings: (i) the standard XMAP that used the estimated inflation constants ($c_1$ and $c_2$) to correct the confounding bias; (ii) a special case of XMAP forced not to correct the confounding bias by setting $c_1 = c_2 = 1$, denoted as 'XMAP (C=I)'. In this example, the SNP rs2053005 locating at the locus 66.55 Mbp–66.85 Mbp in chromosome 15 was significantly associated ($p$-value < $10^{-6}$) in UKBB GWAS (Fig. 5e), but not significant in both Chinese GWAS and EUR Sibship GWAS (Fig. 5f, g). When UKBB and Chinese cohorts were combined for cross-population fine-mapping, the PIP of rs2053005 was computed to be > 0.8 by SuSiEx, PAINTOR, MsCAVIAR, and XMAP (C=I) without accounting for confounding bias. After correcting for confounding bias, the PIP of this signal dramatically decreased in XMAP with a PIP < 0.05, which suggests that the high PIP of the SNP could have been caused by population stratification (Fig. 5h). To test our assumption, we applied cross-population methods to combine Chinese and EUR Sibship GWASs, both of which are known to be less influenced by population structure. As expected, all methods consistently yielded a low PIP for rs2053005 (Fig. 5i). This observation confirmed our assumption that rs2053005 could be a false positive and XMAP was able to exclude this signal by correcting the confounding bias.

## XMAP enables identification of multiple putative causal signals in fine-mapping

With the confounding bias properly corrected, XMAP's efficient algorithm allows us to produce reliable PIP for identifying multiple putative causal variants in thousands of loci across the whole genome. As summarized in Fig. 6a, with a PIP threshold of 0.5, XMAP identified 55 loci harboring more than one putative causal SNPs of height by combing UKBB and Chinese GWASs, among which 6 loci harbor more than 3 causal SNPs and 2 loci harbor 5 causal variants. With a stringent threshold PIP = 0.9, XMAP identified 15 loci with 2 causal SNPs and 9 loci with 3 causal SNPs. To examine the reliability of putative causal SNPs in loci harboring multiple causal signals, we evaluated the replication rates of these SNPs using the Sibship GWAS. Figure 6b and c compare the replication rates of XMAP and SuSiE using their putative causal SNPs with a PIP threshold of 0.9. For loci with more than one putative causal SNPs, XMAP had the best replication rate (i.e., 24/55 = 43.6% SNPs had $p$-values < $10^{-6}$ and 14/55 = 25.5% SNPs had PIP > 0.1). Although SuSiE and SuSiEx can also identify multiple causal signals (Supplementary Fig. 34), they had lower replication rates than XMAP because they cannot correct for confounding bias. For loci with more than two putative causal SNPs, XMAP outperformed SuSiEx and had similar replication rate with SuSiE applied to EUR GWAS. Although PAINTOR and MsCAVIAR can also integrate cross-population GWASs, they are too time-consuming to analyze all loci on the genome when the number of causal signals are set to be larger than 2. We could only run PAINTOR by setting the number of causal signals to 1 and 2. However, PAINTOR often produced unrealistic PIP for loci containing thousands of variants (Supplementary Figs. 34 and 35). Here, we compared the PIP of SNPs computed by XMAP with SuSiEx, PAINTOR and MsCAVIAR using the locus 130.2 Mbp–130.5 Mbp in chromosome 6 as an example. We first combined the GWASs of UKBB (Fig. 6b) and Chinese (Fig. 6c). Clearly, all compared methods suggest that both rs1415701 and rs6569648 had high probability to be causal (Fig. 6e). To test the robustness of compared methods, we replaced the UKBB GWAS with EUR Sibship GWAS (Fig. 6d) which has smaller sample size but is less influenced by confounding bias, and computed the PIP again (Fig. 6f). Because of the reduced sample size, the PIP of rs6569648 computed by MsCAVIAR reduced to 0.78; the PIP computed by PAINTOR substantially differed from its previous output. By contrast, only XMAP and SuSiEx consistently produced high PIP for rs1415701 and rs6569648 (PIP > 0.8).

In the main analysis, we set $K = 10$ to allow the detection of multiple causal variants. The setting $K = 10$ was supported by the analysis

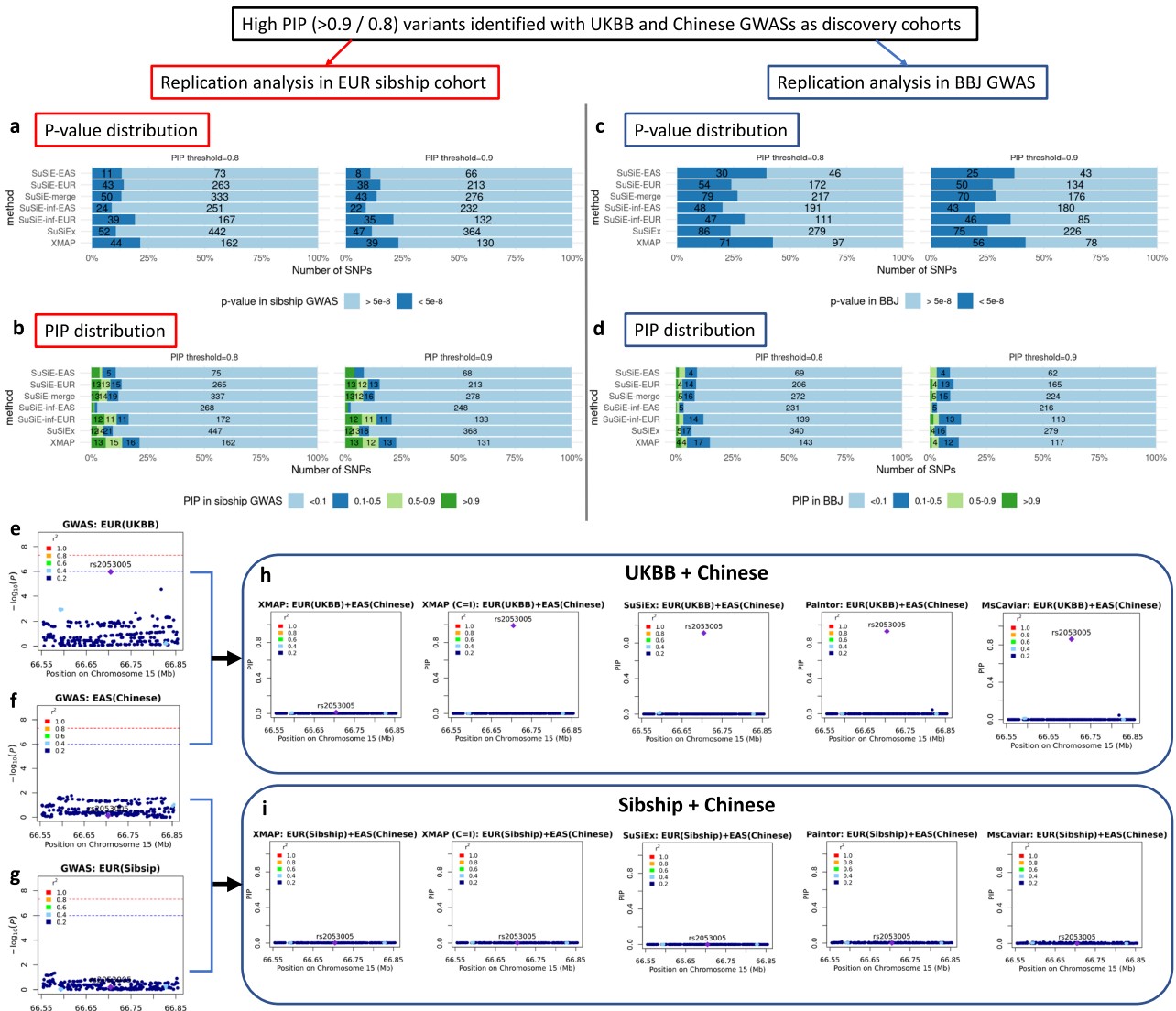

**Fig. 5 | Replication analysis of XMAP and related methods on height GWASs.**
**a–d** Overview of replication analyses of high-PIP fine-mapped SNPs across populations: bar charts showing the fraction and number of fine-mapped SNPs with $p$-value $< 5 \times 10^{-8}$ in the replication cohorts of EUR Sibship GWAS and BBJ cohorts and bar charts showing the distribution of PIP for fine-mapped SNPs computed by SuSiE in the replication cohorts of EUR Sibship GWAS and BBJ. Values for counts < 4 are not shown in the figure. **e–i** Fine-mapping of locus 66.55 Mbp–66.85 Mbp in chromosome 15. The SNP rs2053005 is significant ($p$-value $< 1 \times 10^{-6}$) in UKBB, but not significant in Chinese GWAS and EUR Sibship GWAS. When UKBB and Chinese cohorts were combined for cross-population fine-mapping (**h**), the PIP of

rs2053005 was computed to be > 0.8 by PAINTOR, MsCAVIAR, SuSiEx, and XMAP when we set $c_1 = c_2 = 1$ (XMAP C=I). XMAP estimated the inflation constants of UKBB and BBJ as 1.66 and 1.39, suggesting they are influenced by confounding bias. After correcting for confounding bias, this signal was excluded in XMAP with a PIP < 0.05, which suggests that the high PIP of the SNP could have been induced by uncorrected population stratification. To test our assumption, we combined Chinese and EUR Sibship GWASs, which are both less influenced by confounding factors (both with inflation constant estimated as 1.07). As expected, all methods consistently produced a low PIP for rs2053005 (**i**), which confirmed our assumption and suggested XMAP can reduce spurious signals.

of height as summarized in Fig. 6, where most loci had < 5 causal variants in height. To investigate the sensitivity of fine-mapping performance to the parameter $K$, we further considered $K = 15$ for XMAP and SuSiE. As shown in Supplementary Fig. 32, the number of putative causal SNPs identified by XMAP are highly consistent under different settings of $K$. Besides, the fine-mapped SNPs could be replicated in a consistent rate under different settings of $K$ (Fig. 5a–d and Supplementary Fig. 32). These evidence consolidate our conclusion of the XMAP's robustness to the setting of $K$.

**The XMAP output improves the interpretation of risk variants in their relevant cellular context at single-cell resolution**
Integration of fine-mapping results with single-cell datasets is expected to offer a better interpretation of putative causal variants in their

relevant cellular context at single-cell resolution[6]. However, fine-mapping of an under-presented population often lacks statistical power due to the limited sample size, making the interpretation of causal risk variants difficult. In this section, we show that cross-population fine-mapping results given by XMAP can greatly improve the interpretation of putative causal variants in their relevant cellular context by integrating single-cell datasets. To illustrate this benefit, we carried out SCAVENGE[6] analysis to quantify the enrichment of putative causal variants for 12 blood traits (summarized in Supplementary Table 1) within regions of accessible chromatin using the single-cell assay for transpose-accessible chromatin by sequencing (scATAC-seq). We employed a scATAC-seq dataset that encompasses multiple hematopoietic lineages[36], which includes 33,819 cells from 18 hematological populations (Fig. 7a). Specifically, we have a matrix of

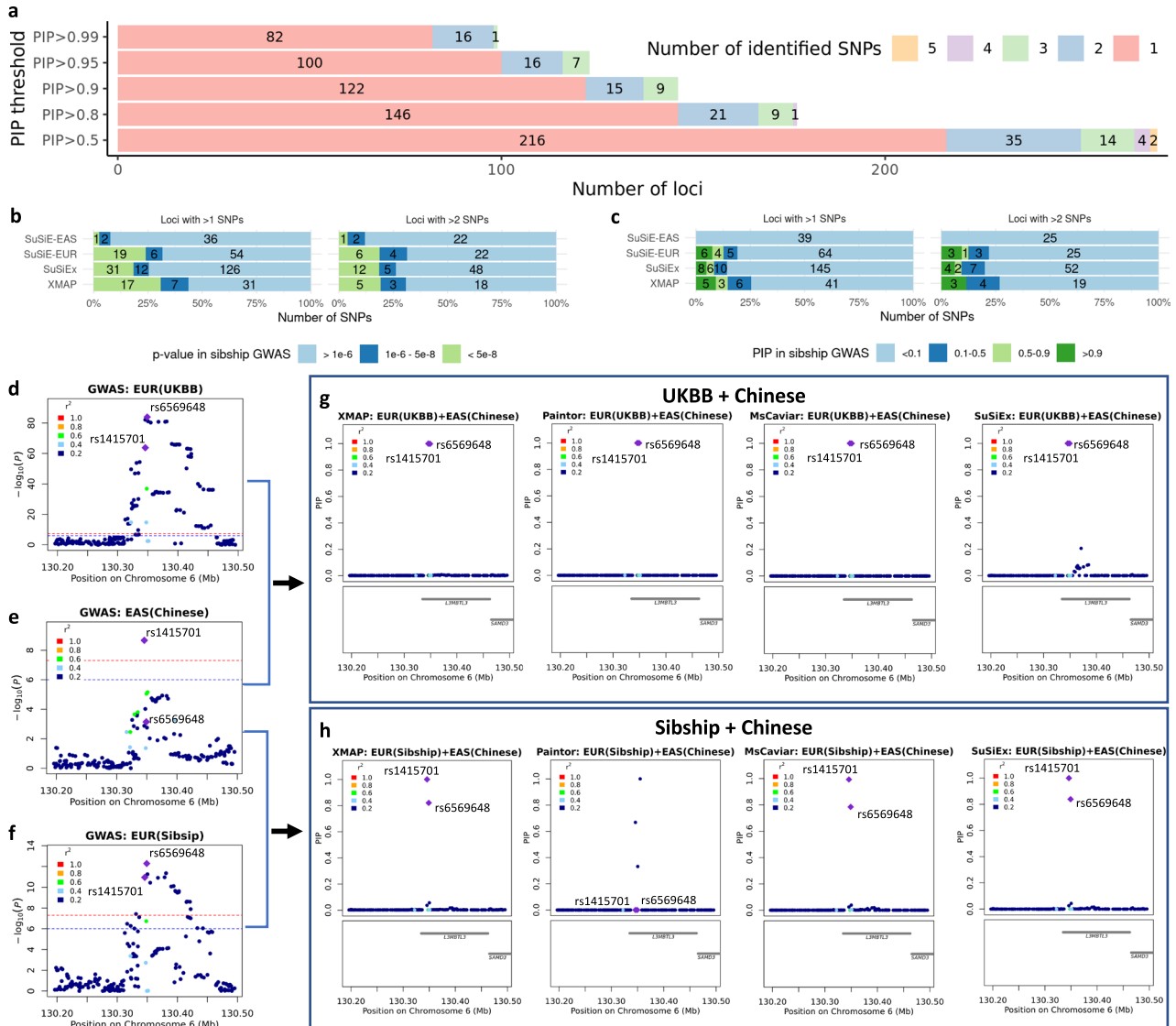

**Fig. 6 | Performance of XMAP in identifying multiple causal variants for height.** **a** Distributions of the number of putative causal SNPs identified by XMAP under different PIP thresholds. **b** With a PIP threshold of 0.9, the *p*-value distributions in the Sibship GWAS replication cohort are shown for putative causal SNPs within loci harboring > 1 and > 2 putative causal SNPs. **c** With a PIP threshold of 0.9, the PIP distributions in the Sibship GWAS replication cohort are shown for putative causal

SNPs within loci harboring > 1 and > 2 putative causal SNPs. **d**–**h** A demonstrative example using the locus 130.2 Mbp–130.5 Mbp in chromosome 6. Manhattan plots of the locus are shown for UKBB GWAS in (**d**), Chinese GWAS in (**e**), and EUR Sibship GWAS in (**f**). The PIP of SNPs in target locus are computed by XMAP, SuSiEx, PAINTOR and MsCAVIAR with GWASs of UKBB+Chinese (**g**) and Sibship+Chinese (**h**).

fragment counts $\mathbf{F} \in \mathbb{R}^{C \times L}$, where $C$ is the number of cells in scATAC-seq data and $L$ is the number of accessible chromatin peaks. To quantify the relevance between the peaks and a phenotype, we first used the XMAP output to compute a vector of weight $\boldsymbol{\eta} \in \mathbb{R}^{L}$ with the $l$-th element of $\boldsymbol{\eta}$ being the sum of XMAP PIP for SNPs within the genomic region of peak $l$, which indicates the relative importance of a peak to the phenotype. The raw cell-trait relevance scores could be computed as $\mathbf{t} = \mathbf{F}\boldsymbol{\eta}$. As such, trait-related cells tend to have larger scores because more causal SNPs are located within their accessible chromatin regions. Then a $Z$-score characterizing the relationship between each pair of cell and trait can be obtained by further correcting for technical confounders, such as GC content bias and PCR amplification, using g-chromVAR[5]. To optimize the inference by leveraging relatedness across individual cells, we constructed a cell-cell similarity network and applied SCAVENGE[6] to assign a trait-relevance score (TRS) for each cell via network propagation. Finally, we simulated null distributions of TRS by using random seed cells for propagation and computed a *p*-value of trait-enrichment for each cell.

The cells with *p*-value < 0.05 were considered as significantly enriched for the trait.

We summarized the identified trait-enriched cells and the median TRS of each cell type in Fig. 7b and Supplementary Fig. 35, respectively. As we can observe, the enriched cells were highly aligned with our knowledge of cell types related to the blood traits. For example, we identified 8388 lymphocyte count (Lym)-related cells, among which 5021 cells were CD4 cells and 2272 were CD8 cells. For traits related to myeloid/compound white cells, including eosinophil count (Eosino), monocyte count (Mono), neutrophil count (Neutro) and white blood cell count (WBC), we observed a substantial number of enriched cells from the CD14+ monocytes. For traits related to red cells such as red blood cell count (RBC), mean corpuscular hemoglobin (MCH), mean corpuscular hemoglobin concentration (MCHC), mean corpuscular volume (MCV), and hemoglobin (HB), a large amount of enriched cells were erythroid cells. These observations indicate that the biological mechanisms of putative causal SNPs identified by XMAP can be interpreted at single-cell resolution. Due to the unbalanced cell type

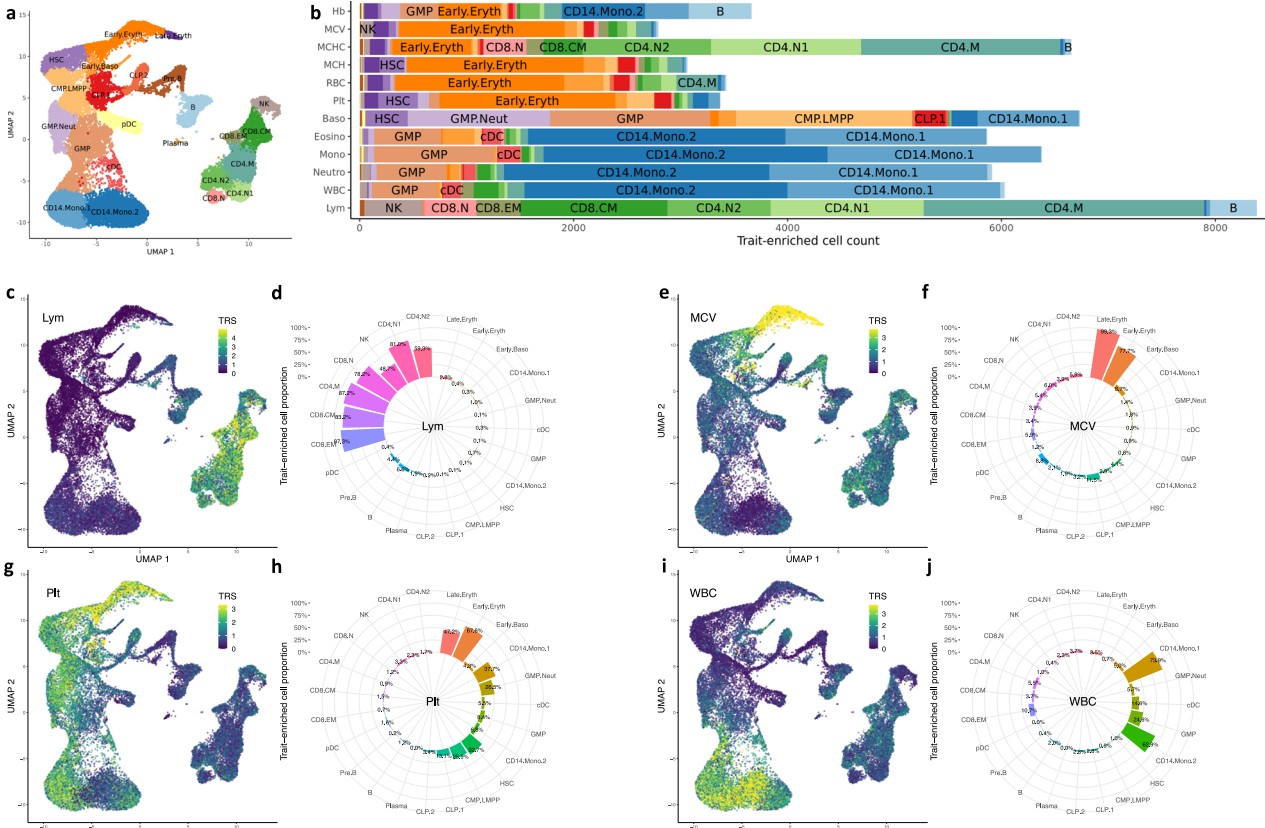

**Fig. 7 | Enrichment of blood cell traits in hematological populations using XMAP fine-mapped SNPs as input. a** The two dimensional uniform manifold approximation and projection (UMAP) plot of scATAC-seq data for 18 hematological populations. **b** The bar plots showing the number of cells significantly enriched in each of the 12 blood traits. The TRS are shown in the UMAP coordinates for four representative traits: Lym (**c**), MCV (**e**), Plt (**g**), and WBC (**i**). The proportions of significantly enriched cells within each population are shown for Lym (**d**), MCV (**f**), Plt (**h**), and WBC (**j**).

composition in the single-cell dataset, cells from rare populations can be under-represented. To rule out the influence of cell type composition on our analysis, we further investigated the proportion of trait-relevant cells within each cell type. We observed that biologically related cell types had largest proportion of enriched cells. For example, MCV was significantly enriched in 99.3% of late stage erythroid cells (Fig. 7e, f), WBC was significantly enriched in more than 60% of CD14⁺ monocytes (Fig. 7i, j), Lym was significantly enriched in a large proportion of CD4 and CD8 cells (Fig. 7c, d), and Plt was significantly enriched in the erythroid cells (Fig. 7g, h). These results suggest that the identification of trait-relevant cells is immune to the cell type composition. As shown in Supplementary Figs. 36–47, we compared the trait-relevant cells obtained by using the XMAP PIP as input with those using the SuSiE PIP from single population analysis as input. Due to the relatively smaller sample size in the BBJ cohort, the trait-relevant cells were less enriched when fine-mapping was performed only using the BBJ GWASs, including GWASs of lymphocyte count (Supplementary Fig. 36 c, d), eosinophil count (Supplementary Fig. 41 c, d), and basophil count (Supplementary Fig. 40c, d). Compared with the single-population fine-mapping result by SuSiE, XMAP can take the advantage of well-powered UKBB GWASs and provide a more accurate fine-mapping result (Supplementary Figs. 36 a, b, 41 a, b, 40 a, b). By integrating with single-cell datasets, the fine-mapping results given by XMAP can offer a better understanding of the putative causal variants in their cellular context at single-cell resolution.

## Discussion

In this paper, we have introduced a novel method named XMAP for cross-population fine-mapping. XMAP is able to improve the statistical

power of fine-mapping by leveraging heterogeneous LD patterns across multiple populations. By correcting the hidden confounding bias in GWAS summary statistics, XMAP can effectively reduce spurious causal signals induced by sample structure. XMAP's fast algorithm allows us to efficiently analyze loci that harbour multiple causal SNPs. Through comprehensive simulations, we showed that XMAP has greater statistical power, better control of false positive rate, and substantially higher computational efficiency for identifying multiple causal signals. We applied XMAP to fine-map causal SNPs of LDL by combining GWASs from EAS, EUR and AFR, achieving substantial gains in statistical power. Furthermore, we showed that XMAP was able to exclude spurious signals and produced reproducible results. By combining the output of XMAP for blood traits with scATAC-seq profiles of hematopoietic cells, we illustrated that the output of XMAP was particularly helpful to characterize the causal mechanism behind phenotypic variation at single-cell resolution. We believe that XMAP can serve as a powerful analytic tool of fine-mapping.

To leverage the genetic diversity from cross-population GWASs, a very recent method, SuSiEx[28], was proposed as a simple extension of SuSiE to the cross-population setting. Although XMAP and SuSiEx have the same goal to leverage genetic diversity by extending the SuSiE framework, XMAP is unique in the following two aspects. First, XMAP introduces an additional genetic component to capture the dense polygenic effects. The inclusion of polygenic effects not only improves the calibration and power of fine-mapping in the presence of non-sparse polygenic effects, it also protects the statistical inference against overfitting when $K$ is specified larger than the ground truth. Second, XMAP can correct confounding bias in GWAS summary statistics. As such, XMAP produces better calibrated PIP in the presence of

confounding bias. This is critical for improving the reproducibility in real data analysis.

Considering the polygenic nature of complex traits, XMAP takes the polygenic effects into account while modeling the major causal effects. The inclusion of polygenic effects benefits fine-mapping in two aspects. First, it captures the small genetic effects, allowing us to focus on the causal SNPs with major genetic impact that can be more biologically interesting for downstream analysis. We verified that the inclusion of the polygenic effects improved the calibration and power in comprehensive simulation studies and offered better reproducibility in real data analyses. Second, the statistical inference of causal effects are protected against over-fitting when $K$ is specified larger than the ground truth. Therefore, we can safely set $K$ to be a larger number, when the ground truth is unknown (Supplementary Fig. 10). A recent study also proposed to account for the polygenic effects in fine-mapping[25]. In particular, SuSiE-inf proposed in this work is similar to XMAP as they both extend SuSiE by incorporating a component of dense polygenic effects in addition to the sparse causal effects captured by SuSiE's sum-of-single-effects models. Despite this similarity, XMAP is different from SuSiE-inf in three aspects. First, XMAP can achieve higher power by leveraging the different LD structures of genetically diverged populations. Second, XMAP corrects confounding bias under the LDSC assumption, which further reduces false positive findings and improves replication rates. Third, XMAP estimates the variance of polygenic effects in a different way than SuSiE-inf. SuSiE-inf uses local SNPs in the target region to estimate model parameters, whose variance highly depends on the GWAS sample size. In contrast, as described in the Method section, XMAP first uses SNPs from the whole-genome to estimate the variance of polygenic effects with LDSC, and then fixes these parameters at their estimates and runs the variational EM algorithm that only updates the parameters of causal effects $\Sigma$. This two-step algorithmic design not only improves the convergence of the variational EM algorithm by reducing the number of parameters but also offers a stable estimate of the polygenic effects. Because SNPs from the entire genome are used for estimation, the parameter estimates of the polygenic component are much more accurate and stable.

Identifying the tissue and cellular context of causal variants is a critical step to understand their biological mechanisms. Existing methods are usually limited to investigation at tissue[37–44] or cell type levels[45–49], which do not fully utilize the rich resources of single-cell profiles. An important feature of XMAP is that it produces outputs that can be integrated with single-cell profiles to illuminate the cellular context of putative causal SNPs at single-cell resolution, offering a unique opportunity to characterize the biological mechanisms across a whole spectrum of cell functions.

Although it is convenient to work with GWAS summary statistics, fine-mapping requires a population-matched reference LD matrix as an input. The inconsistency of LD patterns between reference samples and GWAS samples can lead to false positive findings[24,50–52]. In our main analysis, we have used the in-sample LD references for EAS and UKBB GWAS to minimize the risk of LD mismatching. In practice, if an in-sample LD reference is not available, some diagnostic tools such as SLALOM[51] and DENTIST[50] should be carried out to validate the fine-mapping results and remove suspicious signals.

Our XMAP approach needs more investigation in the following directions. First, similar to PAINTOR and MsCAVAIR, XMAP assumes that the causal variants are shared across populations. Recent studies have reported that some causal signals could be specific to a certain population[53]. Hence, extending XMAP to handle the population-specific causal effects may yield biologically interesting discoveries. Second, causal variants are reported to be distributed disproportionately in the genome, depending on the functional context of the genomic regions[18,26,32,54]. Some recent methods incorporate the information of functional annotation to improve fine-mapping[18,26,32]. It

is interesting to incorporate functional annotations in the causal inference of XMAP, which may further boost the statistical power of fine-mapping. Third, gene-level effects can be more stably shared across populations, as compared to SNP-level effects. A recent study[55] suggests that the correlation of gene-level effects is 20% stronger than SNP-level effects across populations. Therefore, leveraging the genetic diversity at the gene-level for fine-mapping can be also an interesting direction. We will explore these potential extensions in the near future.

## Methods
### The XMAP model
We begin with the probabilistic formulation of XMAP with individual-level GWAS data. For easier introduction, we consider the case of two populations for easier introduction but note XMAP that is generally applicable to analyze multiple populations. Let $\{\mathbf{y}_1,\mathbf{X}_1\}$ and $\{\mathbf{y}_2,\mathbf{X}_2\}$ be the GWAS datasets collected from two different populations, where $\mathbf{y}_1 \in \mathbb{R}^{n_1}$ and $\mathbf{y}_2 \in \mathbb{R}^{n_2}$ are phenotype vectors, $\mathbf{X}_1 \in \mathbb{R}^{n_1 \times p}$ and $\mathbf{X}_2 \in \mathbb{R}^{n_2 \times p}$ are genotype matrices, $p$ is the number of SNPs in the locus of interest, and $n_1$ and $n_2$ are the GWAS sample sizes of populations 1 and 2, respectively. With different recombination rates, the two populations tend to have different LD patterns, i.e., the correlations among columns of $\mathbf{X}_1$ are usually distinct from those of $\mathbf{X}_2$. Without loss of generality, we assume that the columns of $\mathbf{X}_1$ and $\mathbf{X}_2$ have been standardized to have zero mean and unit variance. To relate genotypes and phenotypes, we consider the following linear models:

$$\begin{aligned} \mathbf{y}_1 &= \mathbf{X}_1\mathbf{b}_1 + \mathbf{X}_1\boldsymbol{\phi}_1 + \mathbf{e}_1, \\ \mathbf{y}_2 &= \mathbf{X}_2\mathbf{b}_2 + \mathbf{X}_2\boldsymbol{\phi}_2 + \mathbf{e}_2, \end{aligned} \tag{1}$$

where $\mathbf{b}_1 \in \mathbb{R}^p$ and $\mathbf{b}_2 \in \mathbb{R}^p$ are sparse vectors of causal effects with major impact on phenotypes, $\boldsymbol{\phi}_1 = [\phi_{11},\phi_{12},...,\phi_{1p}]^T \in \mathbb{R}^p$ and $\boldsymbol{\phi}_2 = [\phi_{21},\phi_{22},...,\phi_{2p}]^T \in \mathbb{R}^p$ are dense vectors capturing the polygenic effects[56], and $\mathbf{e}_1 \sim \mathcal{N}(\mathbf{0},\sigma^2_{\mathbf{e}_1}\mathbf{I}_{n_1})$ and $\mathbf{e}_2 \sim \mathcal{N}(\mathbf{0},\sigma^2_{\mathbf{e}_2}\mathbf{I}_{n_2})$ are vectors of independent noises from populations 1 and 2, respectively. Here, we assume that the covariates (e.g., sex, age, and principal components) have been adjusted. The detailed treatment of covariates follows our previous works[30,57]. Unlike previous methods that only consider the overall genetic effects[17–20,22], we separate the genetic effects into causal and polygenic components. This decomposition allows us to focus on the causal SNPs with major genetic impact $\mathbf{b}_1$ and $\mathbf{b}_2$ that can be more biologically interesting for downstream analysis. Accumulating evidence of a shared genetic basis across populations[26,27,30,58,59] implies that $\mathbf{b}_1$ and $\mathbf{b}_2$ tend to have the same set of nonzero entries. Therefore, we expect that the different LD patterns in $\mathbf{X}_1$ and $\mathbf{X}_2$ can be helpful for fine-mapping shared causal SNPs across populations.

To leverage the cross-population GWASs for fine-mapping, we propose to specify model (1) by decomposing the causal genetic effects $\mathbf{b}_1$ and $\mathbf{b}_2$ into $K$ 'single effects':

$$\begin{aligned} \mathbf{y}_1 &= \mathbf{X}_1 \sum_{k=1}^{K} \boldsymbol{\gamma}_k \beta_{1k} + \mathbf{X}_1\boldsymbol{\phi}_1 + \mathbf{e}_1, \\ \mathbf{y}_2 &= \mathbf{X}_2 \sum_{k=1}^{K} \boldsymbol{\gamma}_k \beta_{2k} + \mathbf{X}_2\boldsymbol{\phi}_2 + \mathbf{e}_2, \end{aligned} \tag{2}$$

where $\beta_{1k}$ and $\beta_{2k}$ are effect sizes of the $k$-th causal signal in populations one and two, respectively, $\boldsymbol{\gamma}_k = [\gamma_{k1},...,\gamma_{kp}]^T \in \{0,1\}^p$ in which only one element is 1 and the rest are 0 with $\gamma_{kj} = 1$ indicating the $j$-th variant is responsible for the $k$-th causal signal. This formulation of XMAP has three salient properties. First, through the shared causal status $\boldsymbol{\gamma}_k$, XMAP can leverage the distinct LD patterns between $\mathbf{X}_1$ and $\mathbf{X}_2$. Meanwhile, we allow the two populations to have different effect sizes $\beta_{1k}$ and $\beta_{2k}$. Second, the decomposition of the causal signals into $K$ single causal effects not only allows us to characterize each individual causal signal with an associated credible set[23] but also offers a computational advantage over existing methods, as we shall see later.

Third, the inclusion of the polygenic component also protects the statistical inference against over-fitting when $K$ is specified larger than the ground truth. With this property, we can safely set $K$ to be a reasonably large number, say $K = 10$ by default, when the ground truth is unknown. To infer the causal status $\gamma_k$, we specify the probabilistic structures for the genetic effects in model (2) as follows:

$$\gamma_k \sim \text{Mult}(1, [1/p, \ldots, 1/p]^T),$$
$$\begin{bmatrix} \beta_{1k} \\ \beta_{2k} \end{bmatrix} \sim \mathcal{N}(\mathbf{0}, \boldsymbol{\Sigma}_k), \text{ for } k = 1, \ldots, K, \qquad (3)$$
$$\begin{bmatrix} \phi_{1j} \\ \phi_{2j} \end{bmatrix} \sim \mathcal{N}(\mathbf{0}, \boldsymbol{\Omega}), \text{ for } j = 1, \ldots, p,$$

where $\text{Mult}(1, [1/p, \ldots, 1/p]^T)$ denotes the non-informative categorical distribution of class counts drawn with class probabilities given by $1/p$ for each SNP, $\mathcal{N}(\mathbf{0}, \boldsymbol{\Sigma}_k)$ and $\mathcal{N}(\mathbf{0}, \boldsymbol{\Omega})$ denote the multivariate normal distributions with mean $\mathbf{0}$ and covariance matrices $\boldsymbol{\Sigma}_k = \begin{bmatrix} \sigma_{k1}^2 & \sigma_{k12}^2 \\ \sigma_{k12}^2 & \sigma_{k2}^2 \end{bmatrix}$ and $\boldsymbol{\Omega} = \begin{bmatrix} \omega_1 & \omega_{12} \\ \omega_{12} & \omega_2 \end{bmatrix}$, respectively. The variance components $\boldsymbol{\Sigma} = \{\boldsymbol{\Sigma}_1, \ldots, \boldsymbol{\Sigma}_K\}$ capture the genetic covariance of the two populations attributed to the $K$ causal effects, and $\boldsymbol{\Omega}$ captures the genetic covariance attributed to the polygenic effects.

So far, we have assumed the covariates have been adjusted. In the presence of covariates, we can extend XMAP model in Equation (2) as

$$\mathbf{y}_1 = \mathbf{W}_1 \mathbf{u}_1 + \mathbf{X}_1 \sum_{k=1}^{K} \gamma_k \beta_{1k} + \mathbf{X}_1 \phi_1 + \mathbf{e}_1,$$
$$\mathbf{y}_2 = \mathbf{W}_2 \mathbf{u}_2 + \mathbf{X}_2 \sum_{k=1}^{K} \gamma_k \beta_{2k} + \mathbf{X}_2 \phi_2 + \mathbf{e}_2, \qquad (4)$$

where $\mathbf{W}_1 \in \mathbb{R}^{n_1 \times q_1}$ and $\mathbf{W}_2 \in \mathbb{R}^{n_2 \times q_2}$ are the covariate matrices of populations 1 and 2, respectively, and $\mathbf{u}_1 \in \mathbb{R}^{q_1}$ and $\mathbf{u}_2 \in \mathbb{R}^{q_2}$ are corresponding vectors of covariate effects. To adjust the covariates, we first construct the projection matrices $\mathbf{P}_1 = \mathbf{I} - \mathbf{W}_1(\mathbf{W}_1^T \mathbf{W}_1)^{-1} \mathbf{W}_1^T$ and $\mathbf{P}_2 = \mathbf{I} - \mathbf{W}_2(\mathbf{W}_2^T \mathbf{W}_2)^{-1} \mathbf{W}_2^T$. Then we multiply $\mathbf{P}_1$ on both sides of the first equation and $\mathbf{P}_2$ on both sides of the second equation in model (4). Through this projection, we can obtain a model without covariates

$$\mathbf{y}_1^{\mathbf{P}} = \mathbf{X}_1^{\mathbf{P}} \sum_{k=1}^{K} \gamma_k \beta_{1k} + \mathbf{X}_1^{\mathbf{P}} \phi_1 + \mathbf{e}_1^{\mathbf{P}},$$
$$\mathbf{y}_2^{\mathbf{P}} = \mathbf{X}_2^{\mathbf{P}} \sum_{k=1}^{K} \gamma_k \beta_{2k} + \mathbf{X}_2^{\mathbf{P}} \phi_2 + \mathbf{e}_2^{\mathbf{P}}, \qquad (5)$$

where $\mathbf{y}_1^{\mathbf{P}} = \mathbf{P}_1 \mathbf{y}_1$, $\mathbf{y}_2^{\mathbf{P}} = \mathbf{P}_2 \mathbf{y}_2$, $\mathbf{X}_1^{\mathbf{P}} = \mathbf{P}_1 \mathbf{X}_1$, $\mathbf{X}_2^{\mathbf{P}} = \mathbf{P}_2 \mathbf{X}_2$, $\mathbf{e}_1^{\mathbf{P}} = \mathbf{P}_1 \mathbf{e}_1$, and $\mathbf{e}_2^{\mathbf{P}} = \mathbf{P}_2 \mathbf{e}_2$. As we can observe, model (5) reduces to model (2). With this equivalence, we can work with model (2) without loss of generality.

## The XMAP model for summary-level data
Due to privacy concerns, the individual-level GWAS data may not be easily accessible. Given this situation, we consider the summary-level GWAS data $\{\hat{\mathbf{b}}_1, \hat{\mathbf{s}}_1\} = \{\hat{b}_{1j}, \hat{s}_{1j}\}_{j=1,\ldots,p}$ and $\{\hat{\mathbf{b}}_2, \hat{\mathbf{s}}_2\} = \{\hat{b}_{2j}, \hat{s}_{2j}\}_{j=1,\ldots,p}$ obtained from simple linear regressions:

$$\hat{b}_{1j} = \mathbf{x}_{1j}^T \mathbf{y}_1 / \mathbf{x}_{1j}^T \mathbf{x}_{1j}, \quad \hat{s}_{1j} = \sqrt{\|\mathbf{y}_1 - \mathbf{x}_{1j} \hat{b}_{1j}\|_2^2 / (n_1 \mathbf{x}_{1j}^T \mathbf{x}_{1j})},$$
$$\hat{b}_{2j} = \mathbf{x}_{2j}^T \mathbf{y}_2 / \mathbf{x}_{2j}^T \mathbf{x}_{2j}, \quad \hat{s}_{2j} = \sqrt{\|\mathbf{y}_2 - \mathbf{x}_{2j} \hat{b}_{2j}\|_2^2 / (n_2 \mathbf{x}_{2j}^T \mathbf{x}_{2j})}, \qquad (6)$$

where $\mathbf{x}_{1j} \in \mathbb{R}^p$ and $\mathbf{x}_{2j} \in \mathbb{R}^p$ denote the $j$-th column of $\mathbf{X}_1$ and $\mathbf{X}_2$, respectively. To derive XMAP with summary-level data, we consider the rows of $\mathbf{X}_1$ and $\mathbf{X}_2$ as independently and identically distributed samples drawn from the two populations, respectively. Then, we define the LD matrices $\mathbf{R}_1 = \{r_{1jl}\} \in \mathbb{R}^{p \times p}$ and $\mathbf{R}_2 = \{r_{2jl}\} \in \mathbb{R}^{p \times p}$, where

$r_{1jl} = \mathbb{E}[\mathbf{x}_{1j}^T \mathbf{x}_{1l}/n_1]$ and $r_{2jl} = \mathbb{E}[\mathbf{x}_{2j}^T \mathbf{x}_{2l}/n_2]$ denote the correlation between variants $j$ and $l$ in populations 1 and 2, respectively. We can then obtain the expectation of GWAS effect sizes conditional on $\mathbf{b}$ and $\phi$:

$$\mathbb{E}[\hat{\mathbf{b}}_1 | \mathbf{b}_1, \phi_1] = \mathbb{E}\left[\left(\mathbf{X}_1^T \mathbf{X}_1 (\sum_{k=1}^{K} \gamma_k \beta_{1k} + \phi_1) + \mathbf{X}_1^T \mathbf{e}_1\right)/n_1 | \mathbf{b}_1, \phi_1\right] = \mathbf{R}_1 \sum_{k=1}^{K} \gamma_k \beta_{1k} + \mathbf{R}_1 \phi_1,$$
$$\mathbb{E}[\hat{\mathbf{b}}_2 | \mathbf{b}_2, \phi_2] = \mathbb{E}\left[\left(\mathbf{X}_2^T \mathbf{X}_2 (\sum_{k=1}^{K} \gamma_k \beta_{2k} + \phi_2) + \mathbf{X}_2^T \mathbf{e}_2\right)/n_2 | \mathbf{b}_2, \phi_2\right] = \mathbf{R}_2 \sum_{k=1}^{K} \gamma_k \beta_{2k} + \mathbf{R}_2 \phi_2.$$
$$(7)$$

With this expression, we can connect $\mathbf{b}$ and $\phi$ with GWAS summary data with the following model:

$$\hat{\mathbf{b}}_1 = \mathbf{R}_1 \sum_{k=1}^{K} \gamma_k \beta_{1k} + \mathbf{R}_1 \phi_1 + \epsilon_1, \quad \text{Var}(\epsilon_1) = \hat{\mathbf{S}}_1 \mathbf{R}_1 \hat{\mathbf{S}}_1,$$
$$\hat{\mathbf{b}}_2 = \mathbf{R}_2 \sum_{k=1}^{K} \gamma_k \beta_{2k} + \mathbf{R}_2 \phi_2 + \epsilon_2, \quad \text{Var}(\epsilon_2) = \hat{\mathbf{S}}_2 \mathbf{R}_2 \hat{\mathbf{S}}_2, \qquad (8)$$

where $\hat{\mathbf{S}}_1 \in \mathbb{R}^{p \times p}$ and $\hat{\mathbf{S}}_2 \in \mathbb{R}^{p \times p}$ are diagonal matrices with diagonal terms given as $\{\hat{\mathbf{S}}_1\}_{jj} = \hat{s}_{1j}$ and $\{\hat{\mathbf{S}}_2\}_{jj} = \hat{s}_{2j}$ for $j = 1, \ldots, p$, respectively (see Supplementary Note). To obtain a likelihood function of summary level data, we impose normal distributions for $\hat{\mathbf{b}}_1$ and $\hat{\mathbf{b}}_2$, and Eq. (8) becomes the following model:

$$\hat{\mathbf{b}}_1 \sim \mathcal{N}\left(\mathbf{R}_1 \sum_{k=1}^{K} \gamma_k \beta_{1k} + \mathbf{R}_1 \phi_1, \hat{\mathbf{S}}_1 \mathbf{R}_1 \hat{\mathbf{S}}_1\right),$$
$$\hat{\mathbf{b}}_2 \sim \mathcal{N}\left(\mathbf{R}_2 \sum_{k=1}^{K} \gamma_k \beta_{2k} + \mathbf{R}_2 \phi_2, \hat{\mathbf{S}}_2 \mathbf{R}_2 \hat{\mathbf{S}}_2\right). \qquad (9)$$

Note that model (8) or model (9) is derived by assuming that all the population structures have been properly adjusted in the GWAS summary statistics. To account for the unadjusted confounding bias hidden in GWAS summary statistics, we extend Equation (1) under the genetic drift model of LDSC[33] (see Supplementary Note). We show that model (9) is modified accordingly as

$$\hat{\mathbf{b}}_1 \sim \mathcal{N}\left(\mathbf{R}_1 \sum_{k=1}^{K} \gamma_k \beta_{1k} + \mathbf{R}_1 \phi_1, c_1 \hat{\mathbf{S}}_1 \mathbf{R}_1 \hat{\mathbf{S}}_1\right),$$
$$\hat{\mathbf{b}}_2 \sim \mathcal{N}\left(\mathbf{R}_2 \sum_{k=1}^{K} \gamma_k \beta_{2k} + \mathbf{R}_2 \phi_2, c_2 \hat{\mathbf{S}}_2 \mathbf{R}_2 \hat{\mathbf{S}}_2\right), \qquad (10)$$

where $c_1$ and $c_2$ are LDSC intercepts that indicate the magnitude of inflation in GWAS effect sizes due to confounding bias. In the absence of confounding bias, the values of inflation constants $c_1$ and $c_2$ are close to one. As observed in biobank-scale GWASs[11–13,57], the inflation constant is often larger than one in the presence of confounding bias. These inflation constants in the variance term of model (10) can re-calibrate the GWAS standard error based on the magnitude of confounding effects. The SNP correlation matrices $\mathbf{R} = \{\mathbf{R}_1, \mathbf{R}_2\}$ can be estimated with genotypes either from subsets of GWAS samples or from population-matched reference panels. Under model (3) and (10), we denote the collection of unknown parameters $\theta = \{\boldsymbol{\Sigma}, \boldsymbol{\Omega}, c_1, c_2\}$, and the collections of latent variables $\phi = \{\phi_1, \phi_2\}$, $\gamma = \{\gamma_k\}_{k=1,\ldots,K}$ and $\beta = \{\beta_{1k}, \beta_{2k}\}_{k=1,\ldots,K}$. We shall obtain the parameter estimates $\theta$ and identify causal SNPs with the posterior

$$\Pr(\gamma, \beta, \phi | \hat{\mathbf{b}}, \hat{\mathbf{s}}, \mathbf{R}; \hat{\theta}) = \frac{\Pr(\hat{\mathbf{b}}, \gamma, \beta, \phi | \hat{\mathbf{s}}, \mathbf{R}; \hat{\theta})}{\Pr(\hat{\mathbf{b}} | \hat{\mathbf{s}}, \mathbf{R}; \hat{\theta})}. \qquad (11)$$

## Algorithm and parameter estimation
We adopt a two-step procedure in the XMAP algorithm. In the first step, we apply LDSC to estimate the parameters $c_1$, $c_2$, and $\boldsymbol{\Omega}$ using

summary statistics across the whole genome. This can be achieved based on the LDSC assumption[33]. This step helps to pre-estimate model parameters using genome-wide information and reduces model variance. For $\boldsymbol{\Omega}$, the diagonal terms $\omega_1$ and $\omega_2$ are estimated with the per-SNP heritabilities of the corresponding populations using LDSC. The off-diagonal term $\omega_{12}$ is estimated by the per-SNP co-heritability obtained via bi-variate LDSC. The inflation constants $c_1$ and $c_2$ are estimated by the intercepts of LDSC of the two populations. In the second step, we fix $c_1$, $c_2$, and $\boldsymbol{\Omega}$ at their estimates obtained with LDSC and develop a variational expectation-maximization (VEM) algorithm that only updates $\boldsymbol{\Sigma}$. This algorithmic design not only improves the convergence of the VEM algorithm by reducing the number of parameters but also offers a stable estimate of $\boldsymbol{\Omega}$.

In the following, we describe the VEM algorithm. Traditional maximum likelihood approach estimates $\boldsymbol{\Sigma}$ by maximizing the marginal likelihood

$$\Pr(\hat{\mathbf{b}}|\hat{\mathbf{s}},\mathbf{R};\hat{\boldsymbol{\Omega}},\hat{c}_1,\hat{c}_2,\boldsymbol{\Sigma}) = \sum_{\boldsymbol{\gamma}} \int\int \Pr(\hat{\mathbf{b}}|\hat{\mathbf{s}},\mathbf{R},\boldsymbol{\gamma},\boldsymbol{\beta},\boldsymbol{\phi};\hat{c}_1,\hat{c}_2)\Pr(\boldsymbol{\phi}|\hat{\boldsymbol{\Omega}})\Pr(\boldsymbol{\gamma})\Pr(\boldsymbol{\beta}|\boldsymbol{\Sigma})d\boldsymbol{\beta}d\boldsymbol{\phi}. \tag{12}$$

However, due to the combinatorial nature of $\boldsymbol{\gamma}$, the computational cost for Equation (12) grows exponentially with the number of causal signals $K$. To address this difficulty, we develop an efficient VEM algorithm to estimate $\boldsymbol{\Sigma}$ and approximate the posterior (11). To achieve this, we first derive a lower bound of the logarithm of the marginal likelihood (12)

$$\log \Pr(\hat{\mathbf{b}}|\hat{\mathbf{s}},\mathbf{R};\hat{\boldsymbol{\Omega}},\hat{c}_1,\hat{c}_2,\boldsymbol{\Sigma}) \geq \sum_{\boldsymbol{\gamma}}\int\int q(\boldsymbol{\gamma},\boldsymbol{\beta},\boldsymbol{\phi})\log\frac{\Pr(\hat{\mathbf{b}},\boldsymbol{\gamma},\boldsymbol{\beta},\boldsymbol{\phi}|\hat{\mathbf{s}},\boldsymbol{\Omega},\hat{c}_1,\hat{c}_2,\boldsymbol{\Sigma})}{q(\boldsymbol{\gamma},\boldsymbol{\beta},\boldsymbol{\phi})}d\boldsymbol{\beta}d\boldsymbol{\phi}$$
$$= \mathbb{E}_q[\log \Pr(\hat{\mathbf{b}},\boldsymbol{\gamma},\boldsymbol{\beta},\boldsymbol{\phi}|\hat{\mathbf{s}},\mathbf{R};\hat{\boldsymbol{\Omega}},\hat{c}_1,\hat{c}_2,\boldsymbol{\Sigma}) - \log q(\boldsymbol{\gamma},\boldsymbol{\beta},\boldsymbol{\phi})]$$
$$\equiv \mathcal{L}_q(\boldsymbol{\Sigma}), \tag{13}$$

where the inequality follows Jensen's inequality and $q(\boldsymbol{\gamma},\boldsymbol{\beta},\boldsymbol{\phi})$ is a variational approximation of the posterior (11). For convenience, we denote $\mathbf{b}_{1k}=\gamma_k\beta_{1k}$ and $\mathbf{b}_{2k}=\gamma_k\beta_{2k}$. By leveraging the decomposition in model (2), we propose a factorizable formulation of the variational approximation:

$$q(\boldsymbol{\gamma},\boldsymbol{\beta},\boldsymbol{\phi}) = \prod_{k=1}^{K}q(\mathbf{b}_{1k},\mathbf{b}_{2k})q(\boldsymbol{\phi}) = \prod_{k=1}^{K}q(\boldsymbol{\gamma}_k)q(\beta_{1k},\beta_{2k}|\boldsymbol{\gamma}_k)q(\boldsymbol{\phi}), \tag{14}$$

where $q(\mathbf{b}_{1k},\mathbf{b}_{2k})=q(\boldsymbol{\gamma}_k)q(\beta_{1k},\beta_{2k}|\boldsymbol{\gamma}_k)$ and $q(\boldsymbol{\phi})$ are the distributions of $\{\mathbf{b}_{1k},\mathbf{b}_{2k}\}$ and $\boldsymbol{\phi}$ under the variational approximation, respectively. Unlike previous methods[60,61] that require $\mathbf{b}_{1k}$ and $\mathbf{b}_{2k}$ to be fully factorizable across their $p$ elements, the variational approximation in Equation (14) only requires that $\{\mathbf{b}_{11},\mathbf{b}_{21}\},\ldots,\{\mathbf{b}_{1K},\mathbf{b}_{2K}\}$ are independent and they are independent of $\boldsymbol{\phi}$[23,24], which allows flexible dependencies among the elements of $\mathbf{b}_{1k}$ and $\mathbf{b}_{2k}$. With the above factorizable approximation given by Equation (14), it turns out that both $q(\boldsymbol{\gamma},\boldsymbol{\beta},\boldsymbol{\phi})$ and $\mathcal{L}_q(\boldsymbol{\Sigma})$ can be analytically evaluated. We summarize the VEM algorithm in the following:

**E-step** At the $t$-th iteration, the variational distributions are given as

$$q(\boldsymbol{\gamma}_k|\boldsymbol{\Sigma}^{(t)}) = \text{Mult}(1,\tilde{\boldsymbol{\pi}}_k),$$
$$q\left(\begin{bmatrix}\beta_{1k}\\\beta_{2k}\end{bmatrix}|\gamma_{kj}=1,\boldsymbol{\Sigma}^{(t)}\right) = \mathcal{N}(\tilde{\boldsymbol{\mu}}_{kj},\tilde{\boldsymbol{\Sigma}}_{kj}),$$
$$q\left(\begin{bmatrix}\boldsymbol{\phi}_1\\\boldsymbol{\phi}_2\end{bmatrix}|\boldsymbol{\Sigma}^{(t)}\right) = \mathcal{N}(\tilde{\boldsymbol{\nu}},\tilde{\boldsymbol{\Lambda}}), \tag{15}$$

where $\tilde{\boldsymbol{\pi}}=[\tilde{\pi}_{k1},\ldots,\tilde{\pi}_{kp}]^T\in[0,1]^p$, $\tilde{\boldsymbol{\Sigma}}_{kj}\in\mathbb{R}^{2\times2}$, $\tilde{\boldsymbol{\mu}}_{kj}\in\mathbb{R}^2$, $\tilde{\boldsymbol{\Lambda}}\in\mathbb{R}^{2p\times2p}$, and $\tilde{\boldsymbol{\nu}}\in\mathbb{R}^{2p}$ are variational parameters. The variational parameters are

given as

$$\tilde{\pi}_{kj} = softmax\left(-\log(p) + \tfrac{1}{2}\log|\tilde{\boldsymbol{\Sigma}}_{kj}| + \tfrac{1}{2}\tilde{\boldsymbol{\mu}}_{kj}^T\tilde{\boldsymbol{\Sigma}}_{kj}^{-1}\tilde{\boldsymbol{\mu}}_{kj}\right),$$

$$\tilde{\boldsymbol{\Sigma}}_{kj} = \begin{bmatrix}\tilde{\sigma}_{kj,1}^2 & \tilde{\sigma}_{kj,12}^2\\\tilde{\sigma}_{kj,2}^2 & \tilde{\sigma}_{kj,2}^2\end{bmatrix} = \left(\begin{bmatrix}\frac{r_{1jj}}{\hat{c}_1\hat{s}_{1j}^2} & 0\\0 & \frac{r_{2jj}}{\hat{c}_2\hat{s}_{2j}^2}\end{bmatrix} + (\boldsymbol{\Sigma}_k^{(t)})^{-1}\right)^{-1},$$

$$\tilde{\boldsymbol{\mu}}_{kj} = \begin{bmatrix}\tilde{\mu}_{kj,1}\\\tilde{\mu}_{kj,2}\end{bmatrix} = \tilde{\boldsymbol{\Sigma}}_{kj}\left(\begin{bmatrix}\frac{\hat{\mathbf{b}}_{1j}}{\hat{c}_1\hat{s}_{1j}^2}\\\frac{\hat{\mathbf{b}}_{2j}}{\hat{c}_2\hat{s}_{2j}^2}\end{bmatrix} - \begin{bmatrix}\frac{\mathbf{R}_{1j}^T}{\hat{c}_1\hat{s}_{1j}^2} & 0\\0 & \frac{\mathbf{R}_{2j}^T}{\hat{c}_2\hat{s}_{2j}^2}\end{bmatrix}\left(\sum_{k'\neq1}^{K}\tilde{\boldsymbol{\mu}}_{k'j}\otimes\tilde{\boldsymbol{\pi}}_{k'}+\tilde{\boldsymbol{\nu}}\right)\right),$$

$$\tilde{\boldsymbol{\Lambda}} = \left(\begin{bmatrix}\frac{\hat{\mathbf{S}}_1^{-1}\mathbf{R}_1\hat{\mathbf{S}}_1^{-1}}{\hat{c}_1} & 0\\0 & \frac{\hat{\mathbf{S}}_2^{-1}\mathbf{R}_2\hat{\mathbf{S}}_2^{-1}}{\hat{c}_2}\end{bmatrix}+\hat{\boldsymbol{\Omega}}^{-1}\otimes\mathbf{I}_p\right)^{-1},$$

$$\tilde{\boldsymbol{\nu}} = \tilde{\boldsymbol{\Lambda}}\begin{bmatrix}\frac{\hat{\mathbf{S}}_1^{-2}\hat{\mathbf{b}}_1}{\hat{c}_1}\\\frac{\hat{\mathbf{S}}_2^{-2}\hat{\mathbf{b}}_2}{\hat{c}_2}\end{bmatrix} - \begin{bmatrix}\frac{\hat{\mathbf{S}}_1^{-1}\mathbf{R}_1\hat{\mathbf{S}}_1^{-1}}{\hat{c}_1} & 0\\0 & \frac{\hat{\mathbf{S}}_2^{-1}\mathbf{R}_2\hat{\mathbf{S}}_2^{-1}}{\hat{c}_2}\end{bmatrix}\left(\left(\sum_{k=1}^{K}\tilde{\boldsymbol{\mu}}_{kj}\otimes\tilde{\boldsymbol{\pi}}_k\right)\right), \tag{16}$$

where *softmax* denotes the softmax function to make sure $\sum_{j=1}^{p}\tilde{\pi}_{kj}=1$ and $\otimes$ is the Kronecker product. By combing Equations ((14),(15),(16)), the lower bound (13) can be analytically evaluated as

$$\mathcal{L}_q(\boldsymbol{\Sigma}|\boldsymbol{\Sigma}^{(t)})$$
$$= \left(\sum_k^K\tilde{\boldsymbol{\mu}}_{kj}\otimes\tilde{\boldsymbol{\pi}}_k+\tilde{\boldsymbol{\nu}}\right)^T\begin{bmatrix}\hat{\mathbf{S}}_1^{-2}\hat{\mathbf{b}}_1\\\frac{1}{\hat{c}_1}\\\hat{\mathbf{S}}_2^{-2}\hat{\mathbf{b}}_2\end{bmatrix} - \frac{1}{2}\left(\sum_k^K\tilde{\boldsymbol{\mu}}_{kj}\otimes\tilde{\boldsymbol{\pi}}_k+\tilde{\boldsymbol{\nu}}\right)^T$$
$$\times\begin{bmatrix}\frac{\hat{\mathbf{S}}_1^{-1}\mathbf{R}_1\hat{\mathbf{S}}_1^{-1}}{\hat{c}_1} & 0\\0 & \frac{\hat{\mathbf{S}}_2^{-1}\mathbf{R}_2\hat{\mathbf{S}}_2^{-1}}{\hat{c}_2}\end{bmatrix}\left(\sum_k^K\tilde{\boldsymbol{\mu}}_{kj}\otimes\tilde{\boldsymbol{\pi}}_k+\tilde{\boldsymbol{\nu}}\right)-\sum_j^p\frac{1}{2\hat{c}_1\hat{s}_{1j}^2}r_{1jj}\sum_k^K\tilde{\pi}_{kj}(\tilde{\mu}_{kj,1}^2+\tilde{\sigma}_{kj,1}^2)$$
$$-\sum_j^p\frac{1}{2\hat{c}_2\hat{s}_{2,2j}^2}r_{2jj}\sum_k^K\tilde{\pi}_{kj}(\tilde{\mu}_{kj,2}^2+\tilde{\sigma}_{kj,2}^2)$$
$$+\frac{1}{2}\sum_k^K\left((\tilde{\boldsymbol{\mu}}_{kj}\otimes\tilde{\boldsymbol{\pi}}_k)^T\begin{bmatrix}\frac{\hat{\mathbf{S}}_1^{-1}\mathbf{R}_1\hat{\mathbf{S}}_1^{-1}}{\hat{c}_1} & 0\\0 & \frac{\hat{\mathbf{S}}_2^{-1}\mathbf{R}_2\hat{\mathbf{S}}_2^{-1}}{\hat{c}_2}\end{bmatrix}(\tilde{\boldsymbol{\mu}}_{kj}\otimes\tilde{\boldsymbol{\pi}}_k)\right)$$
$$-\frac{1}{2p}\sum_k\sum_j\text{Tr}(\boldsymbol{\Sigma}_k^{-1}(\tilde{\boldsymbol{\Sigma}}_{kj}+\tilde{\boldsymbol{\mu}}_{kj}\tilde{\boldsymbol{\mu}}_{kj}^T))-\frac{p}{2}\log|2\pi\hat{\boldsymbol{\Omega}}|-\frac{1}{2}\tilde{\boldsymbol{\nu}}^T(\hat{\boldsymbol{\Omega}}^{-1}\otimes\mathbf{I}_p)\tilde{\boldsymbol{\nu}}$$
$$-\frac{1}{2}\text{Tr}\left(\left(\begin{bmatrix}\frac{1}{\hat{c}_1}\hat{\mathbf{S}}_1^{-1}\mathbf{R}_1\hat{\mathbf{S}}_1^{-1} & 0\\0 & \frac{1}{\hat{c}_2}\hat{\mathbf{S}}_2^{-1}\mathbf{R}_2\hat{\mathbf{S}}_2^{-1}\end{bmatrix}+\hat{\boldsymbol{\Omega}}^{-1}\otimes\mathbf{I}_p\right)\tilde{\boldsymbol{\Lambda}}\right)$$
$$+\sum_j^p\sum_k^K\tilde{\pi}_{kj}\log\frac{1}{p}-\sum_j^p\sum_k^K\tilde{\pi}_{kj}\log\tilde{\pi}_{kj}+\frac{1}{2}\sum_j^p\sum_k^K\tilde{\pi}_{kj}(\log|\tilde{\boldsymbol{\Sigma}}_{kj}|-\log|\boldsymbol{\Sigma}_k|)$$
$$+\frac{1}{2}\log|\tilde{\boldsymbol{\Lambda}}|+\text{constant}, \tag{17}$$

where $\text{Tr}(\mathbf{B})$ denotes the trace of the square matrix $\mathbf{B}$, and the constant term does not involve $\boldsymbol{\Sigma}$.

**M-step** We solve $\frac{\partial\mathcal{L}_q}{\partial\boldsymbol{\Sigma}_k}=0$ to obtain the update equation of $\boldsymbol{\Sigma}_k$:

$$\boldsymbol{\Sigma}_k^{(t+1)} = \sum_j^p\tilde{\pi}_{kj}(\tilde{\boldsymbol{\mu}}_{kj}\tilde{\boldsymbol{\mu}}_{kj}^T+\tilde{\boldsymbol{\Sigma}}_{kj}). \tag{18}$$

The above VEM algorithm has computational cost linear to the number of causal variants $K$, allowing for detecting multiple causal effects (e.g., $K=10$) at a given locus.

## Identification of causal variant and construction of credible set
After the convergence of VEM algorithm, we can obtain the approximated posterior probabilities $q(\boldsymbol{\gamma}_k)=\tilde{\boldsymbol{\pi}}_k$, where $\tilde{\pi}_{kj}$ is the posterior

probability that the *k*-th causal signal is contributed by the *j*-th SNP. With the variational approximation given by Equation (15), we can compute the posterior inclusion probability of SNP *j* as

$$PIP_j = \Pr(\gamma_{kj} \neq 0 \text{ for some } k | \hat{\mathbf{b}}, \hat{\mathbf{s}}) \approx 1 - \prod_{k=1}^{K}(1 - \tilde{\pi}_{kj}). \quad (19)$$

We can compute the local false discovery rate of SNP *j* as $fdr_j = 1 - PIP_j$ and prioritize the causal SNPs by controlling the false discovery rate.

The decomposition of causal effects (2) offers an opportunity to characterize the set of SNPs that have high credibility to contribute to an individual causal signal. Let $\mathcal{M} \subset \{1,...,p\}$ be a subset of SNPs from the target locus. A level-*α* credible set of a causal signal *k*, denoted as $CS(k, \alpha)$, is defined as the smallest $\mathcal{M}$ with $\sum_{j \in \mathcal{M}} \tilde{\pi}_{kj} \geq \alpha$. A smaller size of level-*α* credible set (e.g., *α* = 0.9) indicates a higher confidence of the identified causal variants.

### Influence and choice of *K*

The number of causal signals is usually unknown in practice. In XMAP, we do not require *K* to be the number of causal SNPs in the target locus. Instead, because the computational cost of our VEM algorithm only increases linearly with *K*, we can set *K* to a reasonably large number (e.g., *K* = 10) with minor computational overhead. When *K* is larger than the ground truth, the posterior probabilities in the excessive components will be broadly distributed across all SNPs in the locus because there is high uncertainty in the assignment of these causal effects. Importantly, the polygenic component will account for the small genetic effects, forcing the variance of excessive signals toward zero. Therefore, it has very minor influence in prioritization of causal SNPs when including extra causal effects than necessary. To exclude credible sets associated with redundant signal clusters, we follow SuSiE[23] to introduce the purity of credible sets. The purity of a credible set is defined as the smallest absolute correlation between pairs of SNPs within it. In XMAP, we consider the credible sets with purity less than 0.1 in all populations as redundant and discard the associated credible sets.

### Reporting summary

Further information on research design is available in the Nature Portfolio Reporting Summary linked to this article.

## Data availability

Summary statistics of LDL from GLGC can be downloaded at http://csg.sph.umich.edu/willer/public/glgc-lipids2021/. Summary statistics of LDL, height and blood traits from UKBB are available at https://nealelab.github.io/UKBB_ldsc/index.html. Summary statistics of height and blood traits from BBJ are available at http://jenger.riken.jp/en/result. Summary statistics from the within-sibship GWAS are available at https://gwas.mrcieu.ac.uk. The details of publicly available GWAS summary statistics are summarized in Supplementary Table 2. LD files for UKBB British-ancestry and African samples are available at https://data.broadinstitute.org/alkesgroup/UKBB_LD. The scATAC-seq dataset is available at https://github.com/GreenleafLab/MPAL-Single-Cell-2019. The fine-mapping results generated in this study have been deposited in https://github.com/YangLabHKUST/XMAP/tree/main/results.

## Code availability

The XMAP software and source codes in this study were publicly available in GitHub repository of XMAP: https://github.com/YangLabHKUST/XMAP[62]. The software is also provided in the Supplementary Software 1. Example codes for using XMAP are available at https://mxcai.github.io/XMAP-tutorial/Vignettes.html. Relevant codes for reproducing the results of in this study are publicly available at https://github.com/YangLabHKUST/XMAP/tree/main/results.

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

## Acknowledgements

C.Y. was supported by Hong Kong Research Grant Council Grants 16301419, 16308120, 16307221, and 16307322; Hong Kong University of Science and Technology Startup Grants R9405 and Z0428 from the Big Data Institute. M.C. was supported by City University of Hong Kong Startup Grant 7200746. G.C. was supported by National Key Research and Development Project (2021YFF1201200 and 2022YFC3341004) and the National Natural Science Foundation of China (62350004). The computational task for this work was performed by using the X-GPU cluster supported by the Research Grants Council Collaborative Research Fund Grant C6021-19EF.

## Author contributions

M.C. and C.Y. conceived the design of the study and developed the method. M.C. developed the software package, and undertook the statistical and computational analyses with the assistance of Z.W., M.C. and C.Y. wrote the manuscript. Z.W., J.X., X.H., G.C. provided critical feedback during the study and helped revise the manuscript.

## Competing interests

The authors declare no competing interests.
