## [Peer Review File · Nature Communications]

XMAP: Cross-population fine-mapping by leveraging genetic diversity and accounting for confounding biasREVIEWER COMMENTS

Reviewer #1 (Remarks to the Author):

The authors developed a novel statistical method called XMAP for cross-population fine-mapping. They demonstrated the robustness and efficiency of the proposed method by simulations and real data analysis, which addressed some limitations of some existing methods. The manuscript is clear. However, more comprehensive simulations should be done to make the advantages of the proposed method more convincing. The performance of the proposed method on the real data also needs more justifications. Moreover, the model identifiability issue and the validity of the model assumptions should be further discussed. My comments are as follows.

1. In the real data analysis (the second paragraph on page 14), the authors used $PIP > 0.8$ to detect causal SNPs in the discovery datasets, but used a much smaller PIP ($PIP > 0.1$) in the replication datasets. Could the authors explain the reason? What are the replication rates under a larger PIP threshold in the replication cohorts? On the other hand, the replication rates of XMAP are still low (21.4% and 14.9% with $PIP > 0.1$ in two replication cohorts respectively). Does it mean that the confounding bias has not yet been fully accounted for?
2. In line 593 on page 27, the author said that “To ensure the model identifiability, we first apply LDSC to estimate the parameters c_1 , c_2 , and Ω using summary statistics across the whole genome.” I wonder what and why model identifiability issue would occur when estimating these parameters together with Σ simultaneously. Could the authors give more explanations?
3. In the simulation, I wonder why the authors only provide the pAUC with false positive rate < 0.3 . Does XMAP still have better performance than the other methods under different false positive rates?
4. In XMAP, the polygenic effects, the causal effects and random errors are assumed to be normally distributed. In the simulation, the authors only simulated the polygenic effects, the causal effects and random errors from normal distributions, which are under the distribution assumptions of XMAP. I wonder if XMAP can still perform well when the distribution assumptions are violated. How to check the distribution assumptions in practice?

Minor comments

5. (Lines 322 and 327 on page 14) In BBJ replication cohort, the replication rates of XMAP are 21.4% and 14.9%. What is the replication rate of XMAP in EUR Sibship replication cohort?
6. Equation 7 on page 26 is incorrect. Are there any typos here?

Reviewer #2 (Remarks to the Author):

This study proposed a method XMAP which extends the previous fine mapping method SuSiE to multiple populations. The method also integrates the polygenic component and the inflation adjustment in the model. The method is statistically sound, and results are promising as expected given more information is utilized than single population based method. It also demonstrates the usefulness of the identified causal variants in scATAC-seq analysis to identify trait related cell types. The manuscript is well organized and prepared. Overall, the proposed method would be a nice tool to utilize multiple populations to increase the statistical power in fine mapping causal variants. I have the following comments/suggestions that may be helpful for the authors to consider.

1. Recently there is another method SuSiEx available online (<https://www.ncbi.nlm.nih.gov/pmc/articles/PMC9882563/>) which has very similar ideas in extending SuSiE to multiple populations. Can the authors do a comparison with SuSiEx in terms of the statistical modeling and the actual performance on simulated and real data?
2. There has been a very similar method online in terms of adding the polygenic term in the fine mapping model (<https://www.biorxiv.org/content/10.1101/2022.10.21.513123v1>). Please comment on the similarity and differences between the previous model and the polygenic model in this study.
3. Providing a correction using LDSC in a fine mapping model is interesting. However, it is not clear how good the adjustment can achieve. In the simulation results show in Figure 3C, can the authors also add a ground truth as a reference: adjusting PCs in calculating the summary statistics and then applying XMAP. This will give us an idea how good the adjustment can be.
4. SuSiE shows a link between its algorithm to variational approximations and also using EM to estimate the hyperparameters, can the authors comment on the similarity and differences between the SuSiE algorithm and the derivation in this study?
5. If the in-sample LD was used, SuSiE can produce exactly the same results when summary statistics were provided. Does XMAP have the same property? If not, would the results be quite different between individual genotype data and summary statistics when the effect size is large.? The discrepancies between individual genotype data and summary statistics were not considered for early fine mapping methods due to small genotype effects in GWAS.
6. In the simulation studies, please report the calibration results of PIP from XMAP.
7. In the Methods Section when describing representing the causal effects into the sum of single effects, it is better to mention and cite the SuSiE paper there so readers know the source of the idea, because the SuSiE paper provides an extensive justification and illustrates the advantage of this representation.
8. Line 206: typo? Btuy
9. Line 271: the comparison on the number of SNPs from SuSiE is not fair. At least the SNPs identified from each data set should be put together from SuSiE when multiple data sets were used in XMAP.

Responses to Reviewer #1's comments:

1. In the real data analysis (the second paragraph on page 14), the authors used $PIP > 0.8$ to detect causal SNPs in the discovery datasets but used a much smaller PIP ($PIP > 0.1$) in the replication datasets. Could the authors explain the reason? What are the replication rates under a larger PIP threshold in the replication cohorts? On the other hand, the replication rates of XMAP are still low (21.4% and 14.9% with $PIP > 0.1$ in two replication cohorts respectively). Does it mean that the confounding bias has not yet been fully accounted for?

Response:

We thank the reviewer for raising these questions. In Figures 5 B and D of the manuscript, we summarized the number and percentage of putative causal SNPs within four PIP ranges in the replication cohorts: < 0.1 , $0.1 \sim 0.5$, $0.5 \sim 0.9$, and > 0.9 . To observe the replication performance of XMAP under larger PIP thresholds, we have also included these results in Figures R1 and R2 of this letter. As shown in the left panel of Figure R1, among the 206 SNPs identified by XMAP with a PIP threshold of 0.8, 6.3% (13/206) of them had $PIP > 0.9$ and 13.6% of them had $PIP > 0.5$ in the Sibship cohort. As a comparison, with a discovery PIP threshold of 0.8 and replication PIP threshold of 0.9, SuSiE only had replication rates of 4.2% (13/306) among SNPs detected from UKBB GWAS and 3.6%(3/84) among SNPs detected from Chinese cohort. Similar patterns can be observed when a PIP threshold of 0.9 was applied to the discovery cohort (right panel of Figure R1) and when the BBJ cohort was used as a replication GWAS (Figure R2), suggesting XMAP can also achieve the best replication performance under larger PIP thresholds in the replication cohorts.

We agree with the reviewer that the overall replication rates here are not very large. This is also observed in other replication studies of fine-mapping [1]. We believe that the major reason for the low replication rate is the lack of power in the replication study. In our study, we used the Sibship and BBJ cohorts as the replication cohort to minimize the risk of confounding bias. Their sample sizes are 75,030 and 159,095, respectively. GWASs with such limited sample sizes often cannot have enough power to detect causal variants with small effect sizes [2]. This is the reason that we included a small PIP threshold of 0.1 in the analysis. The same PIP threshold is also used in [1] for the investigation of putative causal variants in the replication study. While the replication rate is not a perfect criterion for the aforementioned reason, it offers a way to evaluate the relative performance of fine-mapping in real data, in which we do not observe the true causal variants.

Figure R1. Bar charts showing the distribution of PIP for fine-mapped SNPs computed by SuSiE in the replication cohort of EUR Sibship GWAS. Left: PIP threshold of 0.8 is used to discover causal variants. Right: PIP threshold of 0.9 is used to discover causal variants. Different colors show the PIP ranges of putative causal variants in the replication cohort.

Figure R2. Bar charts showing the distribution of PIP for fine-mapped SNPs computed by SuSiE in the replication cohort of BBJ. Left: PIP threshold of 0.8 is used to discover causal variants. Right: PIP threshold of 0.9 is used to discover causal variants. Different colors show the PIP ranges of putative causal variants in the replication cohort.

2. In line 593 on page 27, the author said that ‘To ensure the model identifiability, we first apply LDSC to estimate the parameters c_1 , c_2 , and Ω using summary statistics across the whole genome.’ I wonder what and why model identifiability issue would occur when estimating these parameters together with Σ simultaneously. Could the authors give more explanations?

Response:

We thank the reviewer for the question. We would like to take this opportunity for clarification. The phrase “To ensure identifiability” means to achieve better convergence of the variational EM algorithm and more stable parameter estimates. We have made the revision to avoid possible confusion.

Let’s briefly example why we prefer to first estimate the parameters c_1 , c_2 , and Ω using summary statistics across the whole genome. To see this, we first derive a variational lower bound of XMAP that updates all parameters $\theta = \{\Sigma, \Omega, c_1, c_2\}$ simultaneously. As described in the manuscript, XMAP aims to obtain the posterior probability

$$\Pr(\gamma, \beta, \phi | \hat{\mathbf{b}}, \hat{\mathbf{s}}, \mathbf{R}; \theta) = \frac{\Pr(\hat{\mathbf{b}}, \gamma, \beta, \phi | \hat{\mathbf{s}}, \mathbf{R}; \theta)}{\Pr(\hat{\mathbf{b}} | \hat{\mathbf{s}}, \mathbf{R}; \theta)}, \tag{R1}$$

where θ is the set of unknown parameters that need to be estimated. When all parameters in θ are unknown, the lower bound of the logarithm of the marginal likelihood is given as follows

$$\begin{aligned} \log \Pr(\hat{\mathbf{b}} | \hat{\mathbf{s}}, \mathbf{R}; \Omega, c_1, c_2, \Sigma) &\geq \sum_{\gamma} \int \int q(\gamma, \beta, \phi) \log \frac{\Pr(\hat{\mathbf{b}}, \gamma, \beta, \phi | \hat{\mathbf{s}}; \Omega, c_1, c_2, \Sigma)}{q(\gamma, \beta, \phi)} d\beta d\phi \\ &= \mathbb{E}_q[\log \Pr(\hat{\mathbf{b}}, \gamma, \beta, \phi | \hat{\mathbf{s}}, \mathbf{R}; \Omega, c_1, c_2, \Sigma) - \log q(\gamma, \beta, \phi)] \\ &\equiv \mathcal{L}_q(\theta), \end{aligned} \tag{R2}$$

where the inequality follows Jensen’s inequality and $q(\gamma, \beta, \phi)$ is a variational approximation of the posterior (R1). With $q(\gamma, \beta, \phi)$ specified in Equation (14) of the manuscript, we can also derive a closed form of Equation (R2) and develop a variational EM algorithm that updates θ at each of its iterations.

While this approach is straightforward, there are two difficulties that make the estimate of θ unstable. First, the variational EM algorithm could easily converge to locally optimal solutions, producing unreliable posterior inclusion probabilities. Second, because a local region usually contains only a

few thousand SNPs, the estimate of θ can have a high variance. To address these difficulties, we first estimate the confounding effects c_1 and c_2 , and the polygenic effects ϕ_1 and ϕ_2 , using summary statistics across the whole genome. This can be achieved based on the LDSC assumption [3]. This step helps to pre-estimate model parameters using genome-wide information and reduces model variance. Therefore, we adopt a two-step estimation procedure as described in the manuscript of XMAP. In the first step, we apply bi-variate LDSC to estimate c_1 , c_2 , and Ω using genome-wide SNPs. In the second step, we fix c_1 , c_2 , and Ω at their estimates obtained with LDSC and run the variational EM algorithm that only updates Σ . This algorithmic design not only improves the convergence of the variational EM algorithm by reducing the number of parameters but also offers a stable estimate of Ω .

To demonstrate the advantage of the two-step procedure in XMAP, we compared the parameter estimates obtained by XMAP and those obtained by SuSiE-inf, a recently proposed fine-mapping method that estimates the variance of the polygenic component within the variational EM algorithm using local SNPs. Because SuSiE-inf only takes a single population GWAS as input, we ran SuSiE-inf with EUR and EAS GWAS separately and focused on the estimates of ω_1 and ω_2 . As shown in Figure R3, the variance of $\hat{\omega}_1$ and $\hat{\omega}_2$ obtained with XMAP are substantially smaller than those obtained by SuSiE-inf, indicating the advantage of XMAP’s algorithmic design. We have also included the comparison of parameter estimates in the revised manuscript and Supplementary Note.

Figure R3. Boxplots showing the estimated ω_1 (EUR) and ω_2 (EAS) obtained by XMAP and SuSiE-inf with simulation data. We varied the sample size of EAS GWAS in 5,000, 10,000, 15,000, and 20,000 and set the sample size of EUR GWAS as 20,000.

3. In the simulation, I wonder why the authors only provide the pAUC with false positive rate < 0.3. Does XMAP still have better performance than the other methods under different false positive rates?

Response:

We thank the reviewer for raising these questions. In Figure R4, we summarize the pAUC from 50 simulation replicates with false positive rate (FPR) thresholds 0.1, 0.2, and 0.3. Here, we set

$K_{true} = 3$, EAS sample size $n_1 = 20,000$, and EUR sample size $n_2 = 20,000$. XMAP had the best performance with false positive rate thresholds 0.2, and 0.3. When $FPR = 0.1$, all methods had similar pAUC. XMAP and PAINTOR were slightly better than single-population methods. A similar pattern can be observed in other settings of sample size and K_{true} , as we summarized in the supplementary Figures 15-17. It is important to note that **pAUC only represents the relative rankings** of true causal SNPs under an empirical FPR threshold. A method can have a high pAUC but low power if the PIP of all SNPs are small. The calibration of PIP and statistical power are more important criteria for evaluating the fine-mapping performance (e.g., the results shown in Figures R5 and R6). Therefore, we mainly focused on the PIP calibration and statistical power in the revised manuscript and included the pAUC under different FPR thresholds in the revised Supplementary Note.

Figure R4. Comparison of pAUC ($FPR < 0.1$) among DAP-G, FINEMAP, SuSiE, PAINTOR, MsCAIVAR, and XMAP across 50 simulations. We set $K_{true} = 3$, EAS sample size $n_1 = 20,000$, and EUR sample size $n_2 = 20,000$ and varied the FPR thresholds $\in \{0.1, 0.2, 0.3\}$.

4. *In XMAP, the polygenic effects, the causal effects and random errors are assumed to be normally distributed. In the simulation, the authors only simulated the polygenic effects, the causal effects and random errors from normal distributions, which are under the distribution assumptions of XMAP. I wonder if XMAP can still perform well when the distribution assumptions are violated. How to check the distribution assumptions in practice?*

Response:

We thank the reviewer for raising this important question. We assume the random errors follow normal distribution because it is a standard practice to perform some transformations on phenotypic values to make them approximately normal. For the effect sizes, it is not easy to check the true distribution in real GWAS data. Instead, we conducted simulation studies to assess the performance of XMAP when the distributions of effect sizes are misspecified. We used the same real genotypes from the EUR and EAS cohorts and focused on the same region with $p = 500$ SNPs as described in the manuscript. The candidate causal SNPs were selected based on the criterion described in

the original simulation. We set $K_{true} = 3$, EAS sample size $n_1 = 20,000$, and EUR sample size as $n_2 = 20,000$. To mimic the violation of XMAP assumptions on effect sizes, we considered scaled t-distribution denoted as $t(df, s)$, where df and s are degrees of freedom and variance of the scaled t-distribution, respectively. The parameter df allows us to specify the discrepancy from the normal distribution and the parameter s controls the strength of genetic effects. For the polygenic effects, we simulated the effect sizes with $\phi_{1j} \sim t(df, 0.005/500)$ and $\phi_{2j} \sim t(df, 0.005/500)$ for $j = 1, \dots, 500$, where 0.005 is the total heritability contributed by polygenic effects of the 500 SNPs, with a per-SNP heritability 10^{-5} . For causal effects, we simulated the effect sizes with $\beta_{1k} \sim t(df, 0.25/500)$ and $\beta_{2k} \sim t(df, 0.25/500)$, which means that each causal SNP has a $0.25/0.005 = 50$ fold per-SNP heritability enrichment compared to non-causal SNPs. We considered two df settings: $df = 4$ and $df = 16$. We compared the performance of fine-mapping across 8 methods, including XMAP, XMAP with $\Omega = \mathbf{0}$, DAP-G, FINEMAP, SuSiE, PAINTOR, and two additional methods suggested by Reviewer 2: SuSiEx and SuSiE-inf. Please refer to the responses to Question 1 and Question 2 of Reviewer 2 for more details. MsCAVIAR was excluded because it is too time-consuming in the setting of $K_{true} = 3$. For each setting, we generated 50 replicates and identified causal variants by controlling the global false discovery rate (FDR). To control the global FDR, we first compute the local false discovery rate of each SNP as $fdr_j = 1 - \text{PIP}_j$. Then we sort SNPs by fdr in ascending order and regard the j -th re-ordered SNP as a risk SNP if $FDR_{(j)} = \frac{\sum_{i=1}^j fdr_{(i)}}{j} < \xi$, where $fdr_{(i)}$ is the i -th ordered fdr , $FDR_{(j)}$ is the corresponding global FDR, and ξ is the selected threshold that the global FDR is controlled at (e.g., 0.1).

When $df = 16$ (Figure R5), XMAP had the best power and calibrated empirical FDR with stringent FDR thresholds ($\text{FDR} \leq 0.2$). With a slightly conservative PIP when $\text{FDR} \geq 0.3$, it still had the highest statistical power among compared methods. SuSiEx and XMAP with $\Omega = \mathbf{0}$ were inflated because they do not account for the polygenic effects. SuSiE and SuSiE-inf were deflated because they only used GWAS from a single population. DAP-G, FINEMAP, and PAINTOR had satisfactory performance in FDR control with stringent FDR thresholds but low statistical power. When $df = 4$, the true distribution of effect sizes is substantially different from the normal distribution (Figure R6). In this setting, XMAP could still achieve good calibration of FDR and high statistical power. Although DAP-G had high power with a less stringent FDR threshold (≥ 0.3), it had an inflated empirical FDR in these settings. Overall, XMAP performed reasonably well in controlling false positives while achieving high statistical power when the effect size distribution is misspecified. This evidence suggests that XMAP is robust and reliable to misspecified effect size distributions in real GWAS data. The simulation results of these settings have been included in the revised manuscript and Supplementary Note.

5. (Lines 322 and 327 on page 14) In BBJ replication cohort, the replication rates of XMAP are 21.4% and 14.9%. What is the replication rate of XMAP in EUR Sibship replication cohort?

Response:

Thank the reviewer for this question. We are sorry for the typo in Line 322 “in the BBJ replication cohort”, which should be “in the Sibship replication cohort”. The replication rate of XMAP in EUR Sibship cohort is 21.4%. We have corrected this typo in the revised manuscript.

6. Equation 7 on page 26 is incorrect. Are there any typos here?

Response:

Thank the reviewer for capturing the typo. A parenthesis was missing in Equation 7. We have

corrected the typo in the revised manuscript.

Figure R5. Comparison of FDR control (top panel) and statistical power (bottom panel) when the causal effects and polygenic effects are generated from a scaled t -distribution with degrees of freedom $df = 16$. In the top panel, the x-axis represents the expected FDR level (ξ) and the y-axis represents the empirical FDR.

Figure R6. Comparison of FDR control (top panel) and statistical power (bottom panel) when the causal effects and polygenic effects are generated from a scaled t -distribution with degrees of freedom $df = 4$. In the top panel, the x-axis represents the expected FDR level (ξ) and the y-axis represents the empirical FDR.

Responses to Reviewer #2’s comments:

1. Recently there is another method *SuSiEx* available online (<https://www.ncbi.nlm.nih.gov/pmc/articles/PMC9882563/>) which has very similar ideas in extending *SuSiE* to multiple populations. Can the authors do a comparison with *SuSiEx* in terms of the statistical modeling and the actual performance on simulated and real data?

Response: We thank the reviewer for providing this important reference. *SuSiEx* is a recently proposed method for cross-population fine-mapping. For ease of presentation, we first write down the statistical model of *SuSiEx* for two populations:

$$\mathbf{y}_1 = \mathbf{X}_1 \sum_{k=1}^K \gamma_k \beta_{1k} + \mathbf{e}_1, \quad \mathbf{y}_2 = \mathbf{X}_2 \sum_{k=1}^K \gamma_k \beta_{2k} + \mathbf{e}_2, \quad (\text{R3})$$

$$\gamma_k \sim \text{Mult}(1, [1/p, \dots, 1/p]^T), \quad \beta_{1k} \sim \mathcal{N}(0, \sigma_{k1}^2), \quad \beta_{2k} \sim \mathcal{N}(0, \sigma_{k2}^2), \quad \text{for } k = 1, \dots, K,$$

where $\mathbf{y}_1 \in \mathbb{R}^{n_1}$ and $\mathbf{y}_2 \in \mathbb{R}^{n_2}$ are two phenotype vectors, $\mathbf{X}_1 \in \mathbb{R}^{n_1 \times p}$ and $\mathbf{X}_2 \in \mathbb{R}^{n_2 \times p}$ are standardized genotype matrices, and $\mathbf{e}_1 \sim \mathcal{N}(\mathbf{0}, \sigma_{\mathbf{e}_1}^2 \mathbf{I}_{n_1})$ and $\mathbf{e}_2 \sim \mathcal{N}(\mathbf{0}, \sigma_{\mathbf{e}_2}^2 \mathbf{I}_{n_2})$ are vectors of independent noises from populations 1 and 2, respectively. *SuSiEx* assumes that the genetic effects are totally contributed by a sum of K causal effects β_1, \dots, β_K . The position of non-zero effects for the k -th causal effect is determined by the binary vector γ_k . From Equation (R3), we can see that *SuSiEx* is a straightforward extension of *SuSiE* to the cross-population setting. It assumes that the causal SNPs (indicated by γ_k) are shared across populations while allowing the effects sizes (β_{1k} and β_{2k}) to be different across populations. Although XMAP and *SuSiEx* have the same goal to leverage genetic diversity by extending the *SuSiE* framework, XMAP is unique in the following two aspects.

- XMAP introduces an additional genetic component to capture the dense polygenic effects. The inclusion of polygenic effects has two benefits. First, it improves the calibration and power of fine-mapping in the presence of non-sparse polygenic effects. Second, it protects the statistical inference against overfitting when K is specified as larger than the ground truth.
- XMAP can correct confounding bias in GWAS summary statistics. As such, XMAP produces better calibrated PIP in the presence of confounding bias. This is critical for improving the replication rate.

These properties allow XMAP to effectively reduce false positive findings and improve the credibility of putative causal SNPs in real GWAS data. In the following, we demonstrate these advantages of XMAP using simulation studies and replication analysis with real GWAS data.

We compare *SuSiEx* and XMAP with simulated GWAS data in two scenarios. In the first scenario, we simulated GWAS summary data with both causal and polygenic effects but without confounding effects. We selected causal SNPs and generated GWAS summary data by following the procedure described in the manuscript. We set the EUR sample size $n_2 = 20,000$, and varied the EAS sample size $n_1 \in \{10,000, 15,000, 20,000\}$ and $K_{true} \in \{1, 2, 3\}$. We set $K = 5$ for *SuSiEx* and XMAP. For each setting, we generated 50 replicates and identified causal variants by controlling the global false discovery rate (FDR). To control the global FDR, we first compute the local false discovery rate of each SNP as $fdr_j = 1 - \text{PIP}_j$. Then we sort SNPs by local FDR in ascending order and regard the j -th re-ordered SNP as a risk SNP if $FDR_{(j)} = \frac{\sum_{i=1}^j fdr_{(i)}}{j} < \xi$, where $fdr_{(i)}$ is the i -th ordered local FDR, $FDR_{(j)}$ is the corresponding global FDR, and ξ is the selected threshold that the global FDR is controlled at a given threshold (e.g., 0.1). We also included a special case of

XMAP by setting $\boldsymbol{\Omega} = \mathbf{0}$, which is equivalent to SuSiEx when there is no confounding bias. As observed in Figure (R7), without accounting for the polygenic effects, SuSiEx and XMAP with $\boldsymbol{\Omega} = \mathbf{0}$ are inflated with very a similar pattern, suggesting they produced many false positives. In contrast, XMAP not only produced well-calibrated PIP in all settings but also achieved higher or comparable statistical power as compared to SuSiEx. To better understand these results, we compared the variance of causal effects (σ_{k1}^2 and σ_{k2}^2 for $k = 1, \dots, 5$) estimated by XMAP and XMAP ($\boldsymbol{\Omega} = \mathbf{0}$). We did not include SuSiEx in this comparison because it does not output $\hat{\sigma}_{k1}^2$ and $\hat{\sigma}_{k2}^2$. However, we expect XMAP ($\boldsymbol{\Omega} = \mathbf{0}$) to have similar performance with SuSiEx because they have the same statistical model and similar calibration and power. For ease of demonstration, we considered the case of only one causal effect, i.e., $K_{true} = 1$, and sorted $\hat{\sigma}_{k1}^2$ and $\hat{\sigma}_{k2}^2$ in decreasing order, respectively. As such, the true causal effect with variance 0.25/500 should be captured by the first causal component in the model. The remaining four causal components in the model are redundant with true variance of zero. We summarized $\hat{\sigma}_{k1}^2$ and $\hat{\sigma}_{k2}^2$ estimated by XMAP and XMAP ($\boldsymbol{\Omega} = \mathbf{0}$) in Figure R9. Although we set $K = 5 > K_{true}$ here, XMAP produced unbiased estimate of σ_{k1}^2 and σ_{k2}^2 by taking the polygenic effects into account. In contrast, without modeling the polygenic effects, XMAP ($\boldsymbol{\Omega} = \mathbf{0}$) overestimated σ_{k1} and σ_{k2} , making its PIP inflated.

In the second scenario, we applied SuSiEx to simulated GWAS data with confounding bias but without polygenic effects. We considered the same settings of n_1 , n_2 , and K_{true} as described above and generated 50 replicates for each setting. As a benchmark, we included XMAP with $c_1 = c_2 = 1$, denoted as XMAP ($\mathbf{C} = \mathbf{I}$), which is equivalent to SuSiEx when there are no polygenic effects. Following the same FDR control procedure to identify causal SNPs, we computed the empirical FDR and statistical power by aggregating across all settings and replicates. As shown in Figure R10, although SuSiEx and XMAP ($\mathbf{C} = \mathbf{I}$) had high statistical power, they were severely inflated due to confounding bias. As a comparison, XMAP performed reasonably well in FDR control, suggesting the confounding bias was effectively corrected.

Lastly, we compare the performance of SuSiEx and XMAP in real GWAS data. We applied SuSiEx to the height GWASs from the UK Biobank (UKBB) cohort and the Chinese cohort as described in the manuscript and evaluated its replication rates using the EUR Sibship and BBJ GWAS cohorts (Figure R11). As expected, XMAP had higher replications rates than SuSiEx in both replication cohorts. For example, when a PIP threshold of 0.8 was applied to the discovery cohorts (top left panel in Figure R11), only 9.5% (47/494) putative causal SNPs achieved a $PIP > 0.1$ in the SibShip replication cohort, which is substantially lower than XMAP (21.4%). Similar patterns can be observed when a PIP threshold of 0.9 was applied to the discovery cohort (top right panel in Figure R11) and when the BBJ cohort was used as a replication GWAS (bottom panels in Figure R11). We have also included two methods in Figure R11: SuSiE-inf and SuSiE-merge, which will be discussed in Question 2 and Question 9. A similar pattern can be observed in the replication analysis of LDL, which we summarized in Supplementary Figure 30. These results have been included in the revised manuscript and Supplementary Note.

Figure R7. Comparison of FDR control among XMAP, SuSiEx, and XMAP with $\Omega = \mathbf{0}$ in the presence of polygenic effects. The x-axis represents the expected FDR level and the y-axis represents the empirical FDR. Clearly, the FDR of SuSiEx and XMAP with $\Omega = \mathbf{0}$ is **not well calibrated** in the presence of polygenic effects.

Figure R8. Comparison of statistical power among XMAP, SuSiEx, and XMAP with $\Omega = \mathbf{0}$ in the presence of polygenic effects. Although SuSiEx seems to have higher power in some cases, we should note that SuSiEx has inflated FDR in all of the above settings (Figure R7).

Figure R9. Comparison of causal effects variance estimated by XMAP and XMAP with $\Omega = \mathbf{0}$ in the presence of polygenic effects. We consider $K_{true} = 1$ here. We sort $\hat{\sigma}_{k_1}^2$ and $\hat{\sigma}_{k_1}^2$ for $k = 1, \dots, 5$ in decreasing order, respectively. Each column represents a causal effect in the model, with decreasing estimated variance from left to right. Each row represents a setting of EAS sample size. The dashed lines represent the true variance of causal effects, which takes value of $0.25/500$ for the first column and 0 for the remaining four columns.

Figure R10. Comparisons of FDR control (top panel) and of statistical power (bottom panel) among XMAP, SuSiEx, and XMAP with $\mathbf{C} = \mathbf{I}$ in the presence of confounding bias. Clearly, the FDR of SuSiEx and XMAP with $\mathbf{C} = \mathbf{I}$ is **not well calibrated** here.

Figure R11. Replication analysis of XMAP, SuSiE, SuSiE-inf, and SuSiEx on height GWAS.

2. There has been a very similar method online in terms of adding the polygenic term in the fine mapping model (<https://www.biorxiv.org/content/10.1101/2022.10.21.513123v1>). Please comment on the similarity and differences between the previous model and the polygenic model in this study.

Response:

We thank the reviewer for providing this important reference. This work proposed to improve fine-mapping by taking the polygenic effects into account. In particular, SuSiE-inf proposed in this work and XMAP both extend SuSiE by incorporating a component of dense polygenic effects in addition to the sparse causal effects captured by SuSiE's sum-of-single-effects models. Despite this similarity, XMAP is different from SuSiE-inf in three aspects.

- First, XMAP can achieve higher power by leveraging the different LD structures of genetically diverged populations.
- Second, XMAP corrects confounding bias under the LDSC assumption, which further reduces false positive findings and improves replication rates.
- Third, XMAP estimates the variance of polygenic effects in a different way than SuSiE-inf. SuSiE-inf uses local SNPs in the target region to estimate model parameters, whose variance highly depends on the GWAS sample size. In contrast, as discussed in the response to Question 2 of Reviewer 1, XMAP first uses SNPs from the whole-genome to estimate the variance of polygenic effects with LDSC, and then fixes these parameters at their estimates and runs the variational EM algorithm that only updates the parameters of causal effects Σ . This two-step algorithmic design not only improves the convergence of the variational EM algorithm by reducing the number of parameters but also offers a stable estimate of the polygenic effects.

In the following, we compare XMAP and SuSiE-inf with a comprehensive simulation study. The settings of the simulation design are described in the response to Question 1. We first evaluate the performance under the scenario with polygenic effects and without confounding bias. We included SuSiE-inf, XMAP, and a special case of XMAP with the polygenic parameters Ω set at their true

values as a reference of the gold standard. As shown in Figure R3, the variance of $\hat{\omega}_1$ and $\hat{\omega}_2$ obtained with XMAP are substantially smaller than those obtained by SuSiE-inf, indicating the benefit of leveraging genome-wide information. Figures R12 and R13 show the performance of FDR control and power, respectively. Clearly, by leveraging genetic diversity, XMAP had better power and FDR calibration than SuSiE-inf. Importantly, by taking advantage of the stable algorithmic design, XMAP achieved similar performance to the gold standard.

Next, we consider the scenario with confounding bias but without polygenic effects. We included three methods for comparison: SuSiE-inf, XMAP, and XMAP with $c_1 = c_2 = 1$, denoted as XMAP ($\mathbf{C} = \mathbf{I}$). As we can observe in Figure R14, SuSiE-inf and XMAP ($\mathbf{C} = \mathbf{I}$) had inflated FDR because the confounding bias was not properly corrected. XMAP performed reasonably well in FDR control, suggesting the confounding bias was effectively corrected.

To benchmark SuSiE-inf in real GWAS data, we applied it to the height GWAS and evaluated the replication rates following the procedure described in the manuscript. As summarized in Figure R11, by taking the polygenic effects into account, SuSiE-inf successfully improved the replication rates of putative causal SNPs from the UKBB EUR cohort, as compared to SuSiE. Interestingly, it had a reduced replication rate when applied to the Chinese EAS cohort. This may be caused by the unstable estimate of polygenic parameters due to the limited sample size of the Chinese GWAS. By contrast, XMAP had a stable performance by incorporating GWASs from genetically diverged populations and correcting the confounding bias through its statistical model. A similar pattern can be observed in the replication analysis of LDL, which we summarized in Supplementary Figure 30. We have included the comparison with SuSiE-inf in the revised manuscript and Supplementary Note.

Figure R12. Comparison of FDR control among XMAP, SuSiE-inf and XMAP with true Ω in the presence of polygenic effects. The x-axis represents the expected FDR level and the y-axis represents the empirical FDR. The FDR of SuSiE-inf is not well calibrated in the presence of polygenic effects.

Figure R13. Comparison of statistical power among XMAP, SuSiE-inf, and XMAP with true Ω in the presence of polygenic effects.

Figure R14. Comparisons of FDR control (top panel) and statistical power (bottom panel) among XMAP, SuSiE-inf, and XMAP with $C = I$ in the presence of confounding bias.

3. *Providing a correction using LDSC in a fine mapping model is interesting. However, it is not clear how good the adjustment can achieve. In the simulation results show in Figure 3C, can the authors also add a ground truth as a reference: adjusting PCs in calculating the summary statistics and then applying XMAP. This will give us an idea how good the adjustment can be.*

Response:

Thank the reviewer for the very constructive comment. Following this suggestion, we first generated GWAS summary statistics by including the PCs as covariates and then applied XMAP to the PC-corrected summary data. As we can observe in the top panel of Figure R15, XMAP produced well-calibrated PIP when the PCs were corrected in the GWAS summary data. When the PCs were not corrected, XMAP was the only method that could still achieve satisfactory FDR control performance. In the bottom panels of Figure R15, we added the ROC of XMAP computed with PC-corrected GWAS data. When the PCs were corrected in GWAS summary data, XMAP had an AUC of 0.863 when the EAS cohort had a sample size of 5,000 and an AUC of 0.888 when the EAS cohort had a sample size of 10,000, which we considered as the highest achievable AUCs. When the PCs were not corrected in GWAS summary data, XMAP could still achieve $> 90\%$ of the optimal AUC. For example, when the EAS cohort had a sample size of 5,000, XMAP achieved 90.8% (0.784/0.863) of the optimal AUC. As a comparison, MsCAVIAR only achieved 88.6% (0.765/0.863) of the optimal AUC. Although XMAP may not fully correct the confounding bias, it effectively reduces the false positive findings under the LDSC assumptions, making it more reliable than existing

methods in the presence of confounding bias. The results of XMAP obtained with PC-corrected GWASs have been included in the revised manuscript.

Figure R15. Top panel: Comparison of FDR control in the presence of confounding bias. The x-axis represents the expected FDR level and the y-axis represents the empirical FDR. Bottom panels: ROC of XMAP, PAINTOR, MsCAVIAR, SuSiEx, SuSiE, SuSiE-inf, FINEMAP, and DAP-G with $K_{true} = 2$, EUR sample size $n_2 = 20,000$, and EAS sample sizes of $n_1 = 5,000$ (bottom left) and $n_1 = 10,000$ (bottom right). XMAP (PC-corrected) represents the results of XMAP obtained with PC-adjusted GWAS data as a reference for optimal performance.

4. *SuSiE* shows a link between its algorithm to variational approximations and also using EM to estimate the hyperparameters, can the authors comment on the similarity and differences between the *SuSiE* algorithm and the derivation in this study?

Response:

We thank the reviewer for the question. *SuSiE* introduces an Iterative Bayesian stepwise selection (IBSS) algorithm to estimate hyperparameters and variational parameters. Similar to the IBSS algorithm, XMAP’s variational EM algorithm aims to maximize the lower bound of log-likelihood by constructing an approximation of the posterior distribution. Despite this similarity, the major differences between the algorithm design of XMAP and *SuSiE* originate from the distinct statistical models. Unlike *SuSiE*, XMAP introduces a component of dense polygenic effects in addition to the sparse causal effects captured by the sum-of-single-effects component. The polygenic effects are controlled by model parameters Ω . Furthermore, under the LDSC assumptions, XMAP captures the confounding bias with the LDSC intercepts c_1 and c_2 . These two unique features of XMAP make its algorithm design different from *SuSiE* in two aspects.

First, unlike the IBSS algorithm that estimates all model parameters simultaneously, XMAP adopts a two-step procedure to reliably and robustly estimate the set of parameters $\{\Omega, c_1, c_2\}$. In the first step, we apply bi-variate LDSC to estimate c_1 , c_2 , and Ω using genome-wide SNPs. In the second step, we fix c_1 , c_2 , and Ω at their estimates obtained with LDSC and run the variational EM algorithm that only updates Σ . This algorithmic design not only improves the convergence of the variational EM algorithm by reducing the number of parameters but also offers a stable estimate of

Ω . Please see more details in the response to Question 2 of Reviewer 1.

Second, the variational approximation of XMAP and SuSiE is different. In the variational approximation to the distribution of latent variables, SuSiE assumes that each single effects are independent (Equation (3.7) in the SuSiE paper). Because XMAP introduces additional polygenic effects $\phi = \{\phi_1, \phi_2\}$ as latent variables, we approximate the posterior distribution of latent variables by assuming that the single effects $\{\mathbf{b}_{11}, \mathbf{b}_{21}\}, \dots, \{\mathbf{b}_{1K}, \mathbf{b}_{2K}\}$ are independent and they are independent of the polygenic effects ϕ , as expressed in Equation (14) of the manuscript. This different treatment of variational approximation leads to a different variational lower bound of log-likelihood for SuSiE and XMAP.

In our simulation analysis, we showed that the two-step procedure under the variational assumption provides an accurate estimate of model parameters (Figure R1). It helps XMAP to stably improve fine-mapping in the presence of polygenic effects and confounding bias.

5. *If the in-sample LD was used, SuSiE can produce exactly the same results when summary statistics were provided. Does XMAP have the same property? If not, would the results be quite different between individual genotype data and summary statistics when the effect size is large? The discrepancies between individual genotype data and summary statistics were not considered for early fine mapping methods due to small genotype effects in GWAS.*

Response: We thank the reviewer for this important question. In the original manuscript, we have made the approximations

$$\begin{aligned}\hat{s}_{1j} &= \sqrt{\|\mathbf{y}_1 - \mathbf{x}_{1j}\hat{b}_{1j}\|_2^2 / (n_1 \mathbf{x}_{1j}^T \mathbf{x}_{1j})} \approx \frac{1}{\sqrt{n_1}}, \\ \hat{s}_{2j} &= \sqrt{\|\mathbf{y}_2 - \mathbf{x}_{2j}\hat{b}_{2j}\|_2^2 / (n_2 \mathbf{x}_{2j}^T \mathbf{x}_{2j})} \approx \frac{1}{\sqrt{n_2}}.\end{aligned}\tag{R4}$$

These approximations implicitly assume that the genetic effect contributed by a single variant is ignorable. This assumption makes the results of summary-level method different from the individual-level method. However, in the special case when the confounding bias is absent, we can derive a summary-level XMAP likelihood that exactly reproduce the individual-level results with in-sample LD. To see this, we first denote the z -scores $\{\hat{\mathbf{z}}_1\}_j = \hat{z}_{1j} = \frac{\hat{b}_{1j}}{\hat{s}_{1j}}$ and $\{\hat{\mathbf{z}}_2\}_j = \hat{z}_{2j} = \frac{\hat{b}_{2j}}{\hat{s}_{2j}}$. We can re-write the XMAP data likelihood in Equation (9) in the main text as distributions of z -scores

$$\begin{aligned}\hat{\mathbf{z}}_1 &\sim \mathcal{N}(\sqrt{n_1} \mathbf{R}_1 (\mathbf{b}_1 + \phi_1), \mathbf{R}_1), \\ \hat{\mathbf{z}}_2 &\sim \mathcal{N}(\sqrt{n_2} \mathbf{R}_2 (\mathbf{b}_2 + \phi_2), \mathbf{R}_2).\end{aligned}\tag{R5}$$

The above model can be modified to remove the assumption of small genetic effects by applying an adjustment to the z -scores:

$$\begin{aligned}\tilde{\mathbf{z}}_1 &:= \mathbf{D}_1^{-1/2} \hat{\mathbf{z}}_1 = \frac{\mathbf{X}_1^T \mathbf{y}_1}{\sqrt{n_1}}, \\ \tilde{\mathbf{z}}_2 &:= \mathbf{D}_2^{-1/2} \hat{\mathbf{z}}_2 = \frac{\mathbf{X}_2^T \mathbf{y}_2}{\sqrt{n_2}},\end{aligned}\tag{R6}$$

where $\mathbf{D}_1 \in \mathbb{R}^{p \times p}$ and $\mathbf{D}_2 \in \mathbb{R}^{p \times p}$ are diagonal matrices with j -th diagonal elements being $\frac{n_1}{n_1 + \hat{z}_{1j}^2}$ and $\frac{n_2}{n_2 + \hat{z}_{2j}^2}$, respectively. These are the maximum likelihood estimates of the residual variance expressed with z -scores. By replacing the $\hat{\mathbf{z}}_1$ and $\hat{\mathbf{z}}_2$ in Equation (R5) with $\tilde{\mathbf{z}}_1$ and $\tilde{\mathbf{z}}_2$, respectively,

we have

$$\begin{aligned}\tilde{\mathbf{z}}_1 &\sim \mathcal{N}(\sqrt{n_1}\mathbf{R}_1(\mathbf{b}_1 + \boldsymbol{\phi}_1), \mathbf{R}_1), \\ \tilde{\mathbf{z}}_2 &\sim \mathcal{N}(\sqrt{n_2}\mathbf{R}_2(\mathbf{b}_2 + \boldsymbol{\phi}_2), \mathbf{R}_2).\end{aligned}\tag{R7}$$

Next, we show that model (R7) is equivalent to the individual-model model. With Equation (1) in the main text, the individual-level XMAP model can be written as

$$\begin{aligned}\mathbf{y}_1 &\sim \mathcal{N}(\mathbf{X}_1(\mathbf{b}_1 + \boldsymbol{\phi}_1), \sigma_{\mathbf{e}_1}^2 \mathbf{I}_{n_1}), \\ \mathbf{y}_2 &\sim \mathcal{N}(\mathbf{X}_2(\mathbf{b}_2 + \boldsymbol{\phi}_2), \sigma_{\mathbf{e}_2}^2 \mathbf{I}_{n_2}).\end{aligned}\tag{R8}$$

By assuming that the phenotype vectors are standardized to have a mean of zero and unit variance and fixing $\sigma_{\mathbf{e}_1}^2 = \mathbf{y}_1^T \mathbf{y}_1 / n_1 = 1$ and $\sigma_{\mathbf{e}_2}^2 = \mathbf{y}_2^T \mathbf{y}_2 / n_2 = 1$, we have

$$\begin{aligned}\frac{\mathbf{X}_1^T \mathbf{y}_1}{\sqrt{n_1}} &\sim \mathcal{N}\left(\frac{\mathbf{X}_1^T \mathbf{X}_1}{n_1} \times \sqrt{n_1}(\mathbf{b}_1 + \boldsymbol{\phi}_1), \frac{\mathbf{X}_1^T \mathbf{X}_1}{n_1}\right), \\ \frac{\mathbf{X}_2^T \mathbf{y}_2}{\sqrt{n_2}} &\sim \mathcal{N}\left(\frac{\mathbf{X}_2^T \mathbf{X}_2}{n_2} \times \sqrt{n_2}(\mathbf{b}_2 + \boldsymbol{\phi}_2), \frac{\mathbf{X}_2^T \mathbf{X}_2}{n_2}\right).\end{aligned}\tag{R9}$$

Clearly, with the substitutions $\tilde{\mathbf{z}}_1 = \frac{\mathbf{X}_1^T \mathbf{y}_1}{\sqrt{n_1}}$, $\tilde{\mathbf{z}}_2 = \frac{\mathbf{X}_2^T \mathbf{y}_2}{\sqrt{n_2}}$, $\mathbf{R}_1 = \frac{\mathbf{X}_1^T \mathbf{X}_1}{n_1}$, and $\mathbf{R}_2 = \frac{\mathbf{X}_2^T \mathbf{X}_2}{n_2}$, model (R9) is equivalent to model (R7). Therefore, by applying the adjustment in Equation (R6), XMAP can produce exactly the same results as the individual-level model. We implement this adjustment in the updated version of XMAP software and provide an option to make such an adjustment. To verify this equivalence, we implemented individual-level XMAP and compared the PIP obtained by both versions of XMAP. We provided the jupyter notebook of this example in the GitHub repository of XMAP (https://github.com/YangLabHKUST/XMAP/blob/main/results/XMAP_adj_z.ipynb) and discussed this adjustment in the revised Supplementary Note.

6. *In the simulation studies, please report the calibration results of PIP from XMAP.*

Response: Thank the reviewer for this suggestion. We use the global false discovery rate (FDR) to evaluate the calibration of PIP in simulation studies. To control the global FDR, we first compute the local false discovery rate of each SNP as $fdr_j = 1 - \text{PIP}_j$. Then we sort SNPs by local FDR in ascending order and regard the j -th re-ordered SNP as a risk SNP if $FDR_{(j)} = \frac{\sum_{i=1}^j fdr_{(i)}}{j} < \xi$, where $fdr_{(i)}$ is the i -th ordered local FDR, $FDR_{(j)}$ is the corresponding global FDR, and ξ is the selected threshold to control the global FDR (e.g., $\xi = 0.1$). We identified putative causal SNPs with a given FDR level and computed the empirical FDR as $1 - \frac{\text{NO. of true SNPs among putative causal SNPs}}{\text{NO. of putative causal SNPs}}$. Because a locus includes at most 3 causal variants in our simulation, we aggregated 50 simulation replicates to improve the precision of empirical FDR. We considered the three scenarios as described in the manuscript: GWAS without polygenic effects and confounding bias (top panel of Figure R16), GWAS with polygenic effects but without confounding bias (middle panel of Figure R16), and GWAS with confounding bias but without polygenic effects (bottom panel of Figure R16). As suggested in Question 9, for each single-population method, we further computed a merged PIP as the larger PIP between the two populations and identified causal SNPs with the ‘merged PIP’. Please see more details in the response to Question 9. The empirical FDR of this ad-hoc method was labeled by the cyan triangle in each panel.

Figure R16. The calibration of PIP under different FDR levels. Top panel: the data were generated without polygenic effects and confounding bias. Middle panel: the data were generated with polygenic effects and without confounding bias. Bottom panel: the data were generated without polygenic effects but with confounding bias. The ad-hoc method by merging PIP of single-population methods across populations is labeled with the cyan triangle in each panel.

In the simulation without polygenic effects and confounding bias (top panel of Figure R16), all the single-population methods performed well without merging PIPs from both populations. All the cross-population methods were also so well-calibrated. In the presence of either polygenic effects (middle panel of Figure R16) or confounding bias (bottom panel of Figure R16), XMAP was the only cross-population method that provided PIP with satisfactory calibration. PAINTOR and MsCAVIAR were inflated in the presence of confounding bias. SuSiEx produced inflated PIP in the presence of either polygenic effects or confounding bias. The analysis of PIP calibration has been included in the revised manuscript.

7. *In the Methods Section when describing representing the causal effects into the sum of single effects, it is better to mention and cite the SuSiE paper there so readers know the source of the idea, because the SuSiE paper provides an extensive justification and illustrates the advantage of this representation.*

Response: We thank the reviewer for this suggestion. We have acknowledged SuSiE for the idea of sum-of-single-effects in the 636-639 lines of the revised manuscript with the sentence:

‘Second, the decomposition of the causal signals into K single causal effects, which comes from the key idea of SuSiE [23], not only allows us to characterize each individual causal signal with an associated credible set but also offers a computational advantage over existing methods, as we shall see later.’

8. *Line 206: typo? Btuy*

Response: Thanks for capturing the typo. We have corrected the typo in the revised manuscript.

9. *Line 271: the comparison on the number of SNPs from SuSiE is not fair. At least the SNPs*

identified from each data set should be put together from SuSiE when multiple data sets were used in XMAP.

Response: We thank the reviewer for this comment. As pointed out by the reviewer, taking the union of the SNPs separately identified in different populations by a single-population method can be a simple way to merge discoveries across different populations. This method is equivalent to the following procedure. First, we applied a single-population method to compute posterior inclusion probability (PIP) of SNP j from different populations separately, denoted as $PIP_{j1}, \dots, PIP_{jT}$, where T is the number of populations. Then, for SNP j , its PIPs are compared across populations, and the largest one is selected to represent the ‘merged PIP’ of the SNP (i.e., $PIP_{j,merge} = \max\{PIP_{j1}, \dots, PIP_{jT}\}$). Next, with the ‘merged PIP’, the FDR control procedure as described in the response to Question 6 is applied to identify causal SNPs. Although this procedure is simple to implement, it involves a step of **taking the maximum of PIP** obtained from multiple populations. Indeed, this post-selection step introduces selection bias and inflates false positive findings. We used both simulation studies and replication analysis with GWAS of LDL and height to illustrate the post-selection bias.

In simulation studies, we evaluated the performance of FDR control by applying this procedure to four single-population methods: DAP-G, FINEMAP, SuSiE, and SuSiE-inf. In Figure R15, we observed that this procedure produced inflated PIP in all three simulation scenarios, especially in the presence of polygenic effects and confounding bias. XAMP was immune to the post-selection bias by incorporating cross-population data through its unified statistical model.

Then, we applied this merging procedure to the PIP of SuSiE in LDL GWAS and evaluated the replication rate using an independent LDL GWAS [4] from the EUR population with a median sample size of 85,785. The details of this replication GWAS data were provided in the revised supplementary text. We computed PIP in this replication cohort using SuSiE and summarized the replication performance in Figure R17. As expected, the merging procedure had a much lower replication rate than XMAP. For example, when GWASs from all three populations were combined, with a PIP threshold of 0.8, 25.5% (38/149) SNPs identified by XMAP had $PIP > 0.1$ in the replication cohort. With the same PIP threshold, only 18.9% (32/109) SNPs identified by the merging procedure had $PIP > 0.1$ in the replication cohort. With a more stringent PIP of 0.99, 31% (31/100) SNPs identified by XMAP had replication $PIP > 0.1$. As a comparison, there were only 18.9% (24/127) SNPs identified by the merging procedure that had $PIP > 0.1$. Similar patterns can be observed in the replication study of height GWAS (Figure R10). Therefore, directly merging the findings of single-population methods is not an optimal way to combine cross-population GWASs because it can introduce post-selection bias. The full results of replication analysis for LDL GWAS has been included in the main text and Supplementary Figure 30.

Figure R17. Replication analysis of XMAP and SuSiE (merge) in LDL GWAS data. The bar charts show the number of fine-mapped SNPs within different PIP ranges of the replication cohort. Each column represents a PIP threshold applied in the discovery cohorts. Each row represents a combination of GWAS populations in the discovery cohorts.

*References

- [1] Masahiro Kanai, Jacob C Ulirsch, Juha Karjalainen, Mitja Kurki, Konrad J Karczewski, Eric Fauman, Qingbo S Wang, Hannah Jacobs, François Aguet, Kristin G Ardlie, et al. Insights from complex trait fine-mapping across diverse populations. *medRxiv*, 2021.
- [2] Daniel McGuire, Yu Jiang, Mengzhen Liu, J Dylan Weissenkampen, Scott Eckert, Lina Yang, Fang Chen, Arthur Berg, Scott Vrieze, et al. Model-based assessment of replicability for genome-wide association meta-analysis. *Nature communications*, 12(1):1964, 2021.
- [3] Brendan K Bulik-Sullivan, Po-Ru Loh, Hilary K Finucane, Stephan Ripke, Jian Yang, Nick Patterson, Mark J Daly, Alkes L Price, and Benjamin M Neale. LD score regression distinguishes confounding from polygenicity in genome-wide association studies. *Nature genetics*, 47(3):291–295, 2015.
- [4] Sarah E Graham, Shoa L Clarke, Kuan-Han H Wu, Stavroula Kanoni, Greg JM Zajac, Shweta Ramdas, Ida Surakka, Ioanna Ntalla, Sailaja Vedantam, Thomas W Winkler, et al. The power of genetic diversity in genome-wide association studies of lipids. *Nature*, 600(7890):675–679, 2021.

REVIEWER COMMENTS

Reviewer #1 (Remarks to the Author):

Minor comments:

1. In Figure 2 of the revised manuscript, there is no subfigure (G), and the caption of (E) is incorrect.
2. On page 31 of the revised manuscript, what does the function $q()$ stand for on the right-hand side in Equation (14)?
3. On page 6 (Figure R5) of the response letter, the powers are all below 0.04 even when the FDR=0.5. There is a dramatic drop in power when the effect size distribution is misspecified (compared with Figure 2(F)). But on page 10 (line 249) of the revised manuscript, the authors said that “XMAP performed reasonably well in controlling false positives while achieving high statistical power”, which does not agree with the results in Figure R5.

Reviewer #2 (Remarks to the Author):

Thanks for the authors' responses. I think my questions are adequately addressed. I only have one additional minor suggestion in the following:

In Figure R15, please also add the PC corrected versions for other methods in the comparison. So readers know the performances of different methods when PCs are used to correct confounding which is commonly used in association analysis.

Responses to Reviewer #1’s comments:

1. In Figure 2 of the revised manuscript, there is no subfigure (G), and the caption of (E) is incorrect.

Response:

Thanks for capturing the typo. We have corrected the typo in the revised manuscript.

2. On page 31 of the revised manuscript, what does the function $q()$ stand for on the right-hand side in Equation (14)?

Response:

We thank the reviewer for raising this questions. We have defined the $q()$ functions in the revised manuscript:

$$q(\boldsymbol{\gamma}, \boldsymbol{\beta}, \boldsymbol{\phi}) = \prod_{k=1}^K q(\mathbf{b}_{1k}, \mathbf{b}_{2k})q(\boldsymbol{\phi}) = \prod_{k=1}^K q(\boldsymbol{\gamma}_k)q(\beta_{1k}, \beta_{2k}|\boldsymbol{\gamma}_k)q(\boldsymbol{\phi}), \quad (\text{R1})$$

where $q(\mathbf{b}_{1k}, \mathbf{b}_{2k}) = q(\boldsymbol{\gamma}_k)q(\beta_{1k}, \beta_{2k}|\boldsymbol{\gamma}_k)$ and $q(\boldsymbol{\phi})$ are the distributions of $\{\mathbf{b}_{1k}, \mathbf{b}_{2k}\}$ and $\boldsymbol{\phi}$ under the variational approximation, respectively. Unlike previous methods [1; 2] that require \mathbf{b}_{1k} and \mathbf{b}_{2k} to be fully factorizable across their p elements, the variational approximation in Equation (R1) only requires that $\{\mathbf{b}_{11}, \mathbf{b}_{21}\}, \dots, \{\mathbf{b}_{1K}, \mathbf{b}_{2K}\}$ are independent and they are independent of $\boldsymbol{\phi}$ [3; 4], which allows flexible dependencies among the elements of \mathbf{b}_{1k} and \mathbf{b}_{2k} .

3. On page 6 (Figure R5) of the response letter, the powers are all below 0.04 even when the $\text{FDR}=0.5$. There is a dramatic drop in power when the effect size distribution is misspecified (compared with Figure 2(F)). But on page 10 (line 249) of the revised manuscript, the authors said that “XMAP performed reasonably well in controlling false positives while achieving high statistical power”, which does not agree with the results in Figure R5.

Response:

We thank the reviewer for identifying this inconsistency. After carefully checking our results, we found that the simulation results used to generate Figure R5 and R6 of the previous response letter were mistakenly used with a much smaller causal effect size, which made all methods produced substantially smaller power than the normal situation. We are sorry for this mistake and would like to update the results. We summarized the simulation results when the effect sizes were generated from scaled t distributions with degrees of freedom 16 and 4 in Figures R1 and R2, respectively. As we can observe from the updated figures, the statistical power is now comparable with the normal scenario in Figure 2 F of the main text. Our conclusions about the scenario of mis-specified effect size distribution are still valid. In both cases, XMAP had the best power and calibrated empirical FDR. SuSiEx and XMAP with $\boldsymbol{\Omega} = \mathbf{0}$ were inflated because they do not account for the polygenic effects. SuSiE and SuSiE-inf were deflated because they only used GWAS from a single population. DAP-G, FINEMAP, and PAINTOR had satisfactory performance in FDR control with stringent FDR thresholds but low statistical power. Although DAP-G had high power with a less stringent FDR threshold (≥ 0.3), it had an inflated empirical FDR in these settings. With these updated results, we can conclude that XMAP performs reasonably well in FDR control while achieving high statistical power when the effect size distribution is mis-specified. The simulation results of these settings have been updated in the revised manuscript and Supplementary

Note. For reproducibility, we also made the codes for generating all simulation results available at <https://github.com/YangLabHKUST/XMAP/tree/main/results/simulation>.

Figure R1. Comparison of FDR control (top panel) and statistical power (bottom panel) when the causal effects and polygenic effects are generated from a scaled t -distribution with degrees of freedom $df = 16$. In the top panel, the x-axis represents the expected FDR level (ξ) and the y-axis represents the empirical FDR.

Figure R2. Comparison of FDR control (top panel) and statistical power (bottom panel) when the causal effects and polygenic effects are generated from a scaled t -distribution with degrees of freedom $df = 4$. In the top panel, the x-axis represents the expected FDR level (ξ) and the y-axis represents the empirical FDR.

Responses to Reviewer #2's comments:

1. In Figure R15, please also add the PC corrected versions for other methods in the comparison. So readers know the performances of different methods when PCs are used to correct confounding which is commonly used in association analysis.

Response:

We thank the reviewer for the suggestion. We show the performance of FDR control for all compared methods when the PC are properly adjusted in GWAS summary statistics (the second row in Figure R3). As expected, most methods are well-calibrated when the PC was properly adjusted. PAINTOR was still inflated because it often produced unrealistically high PIP, which was also reported in a recent study [5]. Although the adjustment of PC can improve calibration, in real GWAS data, many confounding effects, such as socioeconomic status [6] and geographic clustering [7; 8] in GWAS samples, cannot be fully corrected by including PCs as covariates. Our simulation (the first row in Figure R3) suggests that XMAP is the only method that can still yield satisfactory FDR control when the confounding effects are not adjusted in the summary data. The simulation results of these settings have been included in the revised Supplementary Note.

Figure R3. Comparison of FDR control in the presence of confounding bias. The x-axis represents the expected FDR level and the y-axis represents the empirical FDR. The panels in the first row show the FDR obtained with uncorrected PC in summary statistics. The panels in the second row show the results obtained with PC-corrected GWAS summary statistics.

*References

- [1] Yongtao Guan and Matthew Stephens. Bayesian variable selection regression for genome-wide association studies and other large-scale problems. *The Annals of Applied Statistics*, 5(3):1780–1815, 2011.
- [2] Peter Carbonetto and Matthew Stephens. Scalable variational inference for bayesian variable selection in regression, and its accuracy in genetic association studies. *Bayesian analysis*, 7(1):73–108, 2012.
- [3] Gao Wang, Abhishek Sarkar, Peter Carbonetto, Matthew Stephens, et al. A simple new approach to variable selection in regression, with application to genetic fine mapping. *Journal of the Royal Statistical Society Series B*, 82(5):1273–1300, 2020.
- [4] Yuxin Zou, Peter Carbonetto, Gao Wang, and Matthew Stephens. Fine-mapping from summary data with the “sum of single effects” model. *PLoS genetics*, 18(7):e1010299, 2022.
- [5] Ran Cui, Roy A Elzur, Masahiro Kanai, Jacob C Ulirsch, Omer Weissbrod, Mark Daly, Benjamin Neale, Zhou Fan, and Hilary K Finucane. Improving fine-mapping by modeling infinitesimal effects. *bioRxiv*, pages 2022–10, 2022.
- [6] Abdel Abdellaoui, David Hugh-Jones, Loic Yengo, Kathryn E Kemper, Michel G Nivard, Laura Veul, Yan Holtz, Brendan P Zietsch, Timothy M Frayling, Naomi R Wray, et al. Genetic correlates of social stratification in great britain. *Nature human behaviour*, 3(12):1332–1342, 2019.
- [7] Simon Haworth, Ruth Mitchell, Laura Corbin, Kaitlin H Wade, Tom Dudding, Ashley Budu-Aggrey, David Carslake, Gibran Hemani, Lavinia Paternoster, George Davey Smith, et al. Apparent latent structure within the uk biobank sample has implications for epidemiological analysis. *Nature communications*, 10(1):1–9, 2019.
- [8] Abdel Abdellaoui, Conor V Dolan, Karin JH Verweij, and Michel G Nivard. Gene–environment correlations across geographic regions affect genome-wide association studies. *Nature genetics*, 54(9):1345–1354, 2022.

REVIEWERS' COMMENTS

Reviewer #1 (Remarks to the Author):

[none]

Reviewer #2 (Remarks to the Author):

I am glad to see that most methods result in well-calibrated results when PCs are correctly used in the association analysis. The authors might add a discussion on the importance of controlling confounding factors (including enough PCs) in the association analysis stage before fine mapping. For more complex causal modeling, such as those involving socioeconomic and geographic factors, as discussed in Abdellaoui et al. (Nature genetics, 2022), whether to and how to correct for them depend on the goals of the research.

Responses to Reviewer #2's comments:

1. I am glad to see that most methods result in well-calibrated results when PCs are correctly used in the association analysis. The authors might add a discussion on the importance of controlling confounding factors (including enough PCs) in the association analysis stage before fine mapping. For more complex causal modeling, such as those involving socioeconomic and geographic factors, as discussed in Abdellaoui et al. (*Nature genetics*, 2022), whether to and how to correct for them depend on the goals of the research.

Response:

We thank the reviewer for the constructive suggestion. We have added the following discussion in lines 276-281 of the main text:

When the PCs were not corrected, all existing methods had different levels of FDR inflation, suggesting that they requires correcting confounding factors in the association mapping stage before conducting fine-mapping. However, in real GWAS data, many confounding effects, such as socioeconomic status [1] and geographic clustering [2; 3] in GWAS samples, cannot be fully corrected by including PCs as covariates.

*References

- [1] Abdel Abdellaoui, David Hugh-Jones, Loic Yengo, Kathryn E Kemper, Michel G Nivard, Laura Veul, Yan Holtz, Brendan P Zietsch, Timothy M Frayling, Naomi R Wray, et al. Genetic correlates of social stratification in great britain. *Nature human behaviour*, 3(12):1332–1342, 2019.
- [2] Simon Haworth, Ruth Mitchell, Laura Corbin, Kaitlin H Wade, Tom Dudding, Ashley Budu-Aggrey, David Carslake, Gibran Hemani, Lavinia Paternoster, George Davey Smith, et al. Apparent latent structure within the uk biobank sample has implications for epidemiological analysis. *Nature communications*, 10(1):1–9, 2019.
- [3] Abdel Abdellaoui, Conor V Dolan, Karin JH Verweij, and Michel G Nivard. Gene–environment correlations across geographic regions affect genome-wide association studies. *Nature genetics*, 54(9):1345–1354, 2022.